# A methodology to compile multi-hazard interrelationships in a data-scarce setting: an application to the Kathmandu Valley, Nepal

**Harriet E. Thompson**[1], **Joel C. Gill**[2], **Robert Šakić Trogrlić**[3], **Faith E. Taylor**[1], **and Bruce D. Malamud**[4]

[1]Department of Geography, King's College London, London, WC2B 4BG, UK
[2]School of Earth and Environmental Sciences, Cardiff University, Cardiff, CF10 3AT, UK
[3]International Institute for Applied Systems Analysis (IIASA), Laxenburg, A-2361, Austria
[4]Institute of Hazard, Risk and Resilience (IHRR), Durham University, Durham, DH1 3LE, UK

**Correspondence:** Harriet E. Thompson (harriet.e.thompson@kcl.ac.uk)

**Abstract.** This paper introduces a multifaceted methodology to identify and compile single natural hazards and multi-hazard interrelationships within the context of data-scarce urban settings, exemplified by the Kathmandu Valley, Nepal. This approach integrates (i) five blended types of evidence to support a more nuanced and holistic understanding of a hazardscape where data are scarce and (ii) a 2 h practitioner stakeholder workshop with seven participants to provide greater context to the hazards, consider their impacts through the co-production of multi-hazard interrelationship scenarios, and show how this methodology could support more people-centred disaster risk reduction (DRR) strategies. We use blended evidence types, including academic literature, grey literature, media, databases, and social media, to systematically search for exemplars of single hazard types and multi-hazard interrelationships that have influenced or could potentially influence the Kathmandu Valley. We collated 58 sources of evidence for single hazard types and 21 sources of evidence for multi-hazard interrelationships. Using these sources, our study identified 21 single hazard types across 6 hazard groups (geophysical, hydrological, shallow Earth processes, atmospheric, biophysical, and space/celestial hazards) and 83 multi-hazard interrelationships (12 have direct case study evidence of previous influence in the Kathmandu Valley) that might influence the Kathmandu Valley. These exemplars are collated into a Kathmandu Valley Single Hazards and Multi-Hazard Interrelationships Database (Thompson et al., 2024) accompanying this paper. We supplement these exemplars with multi-hazard interrelationship scenarios and multi-hazard impacts developed by practitioner stakeholders engaged in DRR research and practice in the Kathmandu Valley. The results illustrate the complexity of the hazardscape, with many single hazard types and multi-hazard interrelationships potentially influencing the Kathmandu Valley. The research emphasises the importance of inclusive DRR strategies that recognise disaggregated impacts experienced by different social groups. This knowledge can inform the development of dynamic risk scenarios in planning and civil protection, thus strengthening multi-hazard approaches to DRR in "Global South" urban areas such as the Kathmandu Valley.

## 1 Introduction

This paper introduces a multifaceted methodology designed to identify and compile single hazard types and multi-hazard interrelationships in data-scarce urban areas in the "Global South" (noting that we recognise the broader contestations over the term), setting out a rationale and approach to identifying and using exemplars of multi-hazard events to characterise the hazardscape. In this introduction, we present previous work on multi-hazard interrelationships (Sect. 1.1), outline current trends in multi-hazard research (Sect. 1.2), introduce single hazards and multi-hazards in Nepal (Sect. 1.3), and consider multi-hazard data challenges in the Kathmandu Valley context (Sect. 1.4) as the setting to which we apply our methodology.

## 1.1 Previous work on multi-hazard interrelationships

Studies of natural hazards often focus on single, discrete hazards, such as earthquakes, floods, and storms, with more limited knowledge of multi-hazard interrelationships and their impacts (Gill et al., 2020; De Angeli et al., 2022; Ward et al., 2022). Different authors have developed classifications for multi-hazard interrelationships (e.g. Kappes et al., 2010; Duncan, 2014; Van Westen et al., 2014). However, each classification shares many features and typically includes one or more of the following interrelationship types (Gill and Malamud, 2016, 2017; Ciurean et al., 2018; Gill et al., 2022):

- *Compound (or coincident hazards)*. Two or more independent hazards affect the same area spatially and/or temporally.

- *Concurrent or consecutive hazards*. Two or more hazards occur consecutively, resulting in increased stress on a certain area.

  - *Triggering relationships*. One hazard causes another hazard to occur.

  - *Increased-probability relationships*. One hazard increases the magnitude and/or likelihood of further hazards in the future.

- *Catalysis–impedance relationships*. The action of a primary hazard triggers or increases the probability of a secondary hazard.

Examples of multi-hazard interrelationships include an earthquake triggering a landslide, which may block a river and trigger flooding downstream, or drought increasing the probability of wildfire. Single hazard events are often well documented in the literature (e.g. Nehren et al., 2013; World Bank Group and the Asian Development Bank, 2021) and international databases such as EM-DAT (CRED, 2024) and DesInventar Sendai (UNDRR, 2024), particularly concerning events with large magnitudes or where impacts turn into disasters. Databases CE1 detailing the breadth of single hazard types in particular regions are generally rarer in the Global South, yet localised perspectives on multiple single hazards are improving (GFDRR, 2021). Conversely, the systematic knowledge of multi-hazard interrelationships in urban areas in low- and middle-income countries (Šakić Trogrlić, 2024b) and their impacts is still limited. This is particularly the case regarding further characterisation and quantification of multi-hazard interrelationships using a universal framework (Tilloy et al., 2019; Gill et al., 2020; Ward et al., 2022; Hochrainer-Stigler et al., 2023). Table 1 summarises six studies compiling multi-hazard interrelationships on a regional scale, including the systematic methodologies used and the region to which the methodology is applied. The methodologies vary from critical literature reviews to multi-hazard risk analyses as tools to gather information about multi-hazard interrelationships across geographical regions.

## 1.2 Current trends in multi-hazard research

In the past decade, the natural hazard community has evolved towards a more nuanced understanding of multi-hazards, defined as the "(1) selection of multiple major hazards that the country faces, and (2) specific contexts where hazardous events may occur simultaneously, cascadingly or cumulatively over time, and taking into account the potential interrelated effects" (UNDRR, 2017b). There has been increasing awareness of the importance of considering multi-hazards since their inclusion in the Sendai Framework for Disaster Risk Reduction (DRR) 2015-30 (UNDRR, 2015) and implementation of the UNDRR (2017b) definition. Multi-hazard approaches to DRR are an integral part of the vision and research objectives of the Sendai Framework. In Nepal, the focus of this paper, decision-makers and multi-hazard researchers have expressed the need to better integrate multi-hazard approaches into DRR strategies (Government of Nepal, 2018; Gautam et al., 2021), although practical adoption is much more challenging (Aksha et al., 2020). Single hazard events in Nepal are documented in open-access databases such as the Nepal DRR Portal (2024), BIPAD (Building Information Platform Against Disasters) Portal (NDRRMA, 2024), EM-DAT (CRED, 2024), and DesInventar Sendai (UNDRR, 2024). It would be beneficial to have a similarly standardised framework for collecting and recording multi-hazard event data (Tamrakar and Bajracharya, 2020).

A political declaration at the midterm review of the Sendai Framework noted, in Article 8, the increasingly complex nature of disaster risk, considering interrelated impacts across regions and sectors. Article 20 called for improved collection and analysis of hazard, disaster event, and impact data, specifically disaggregated data by social group, e.g. by "income, sex, age and disability" (UNDRR, 2023). The scarcity of multi-hazard event and impact data in some Global South urban areas presents a significant challenge to risk-sensitive DRR strategies (Paudyal et al., 2015; Aksha et al., 2020). Within the context of these urban areas, where the interaction of exposure to multiple hazards and high vulnerability combine to exacerbate risk (Hallegatte et al., 2020; Timsina et al., 2020 TS1), mapping of multi-hazard interrelationships can inform effective and people-centred DRR strategies (Scolobig et al., 2015; UNDRR, 2023). Considering these challenges, the DRR community has identified the need for a greater breadth and depth of multi-hazard data from diverse sources. This call for more nuanced multi-hazard data is a critical priority in better understanding hazardscapes, applied in this paper as a framework to understand the connections between hazards, physical landscapes, socio-political factors and global influences (e.g. Mustafa, 2005; Khan, 2009), and their impacts (Gill et al., 2021b; Šakić Trogrlić et al., 2022; Šakić Trogrlić et al., 2024a). Here, we apply a single hazard and multi-hazard interrelationship scoping methodology using blended evidence types in the context of the Kathmandu

**Table 1.** Summary of six systematic studies compiling multi-hazard interrelationships on a regional scale.

| Publication name | Systematic methodologies | Region methodology is applied to |
| --- | --- | --- |
| "A multi-hazard framework for spatial-temporal impact analysis" (De Angeli et al., 2022) | Critical literature review and stakeholder workshops. The methodology considered types of multi-hazard interrelationships, impacts, and stakeholder perspectives to develop a multi-hazard impact framework. | Po Valley, Italy |
| "Hazard interaction analysis for multi-hazard risk assessment: a systematic classification based on hazard-forming environment" (Liu et al., 2016) | Given different potential multi-hazard interrelationship types, the probability and magnitude of multi-hazards occurring together were calculated using hazard interaction analysis. | Yangtze River Delta, China |
| "Spatial pattern of hazards and hazard interactions in Europe" (Tarvainen et al., 2006) | A causal correlation was used to identify multi-hazard interrelationships and select those which exceeded average hazard intensities for a given region. | Europe |
| "Construction of regional multi-hazard interaction frameworks, with an application to Guatemala" (Gill et al., 2020) | A comprehensive, systematic, and evidenced regional multi-hazard interrelationships framework was populated using internationally accessible literature, locally accessible civil-protection bulletins, field observations, stakeholder interviews, and a stakeholder workshop. | Guatemala (national) and southern highlands of Guatemala (subnational) |
| "From single- to multi-hazard risk analyses: a concept addressing emerging challenges" (Kappes et al., 2010) | Multi-hazard interrelationships were identified using a matrix, modelled, and incorporated into a multi-hazard risk analysis. | Barcelonnette Basin, French Alps |
| "A theoretical model for cascading effects analyses" (Zuccaro et al., 2018) | Development of a cascading effects scenario analysis model, incorporating exposure data and hazard and impact models. This scenario analysis was applied to a hypothetical hazard cascade of an eruption of Nea Kameni volcano, Santorini, Greece. | Santorini, Greece |

Valley, Nepal, as an example of a multi-hazard, data-scarce urban setting.

## 1.3 Single hazards and multi-hazards in Nepal

The global population in urban areas, as of 2022, is estimated to be 4.2 billion people, with the most rapid growth in informal settlements in low- to middle-income countries with lower adaptive capacity (Dodman et al., 2022). Within these countries, Nepal experiences high exposure to multi-hazards, which is coupled with challenges presented by its medium Human Development Index (HDI) of 0.602 in 2021, on a scale of 143 out of 191 countries globally (UNDP, 2022). Nepal has an estimated global Multidimensional Poverty Index (MPI) of 0.074 based on a 2019 survey, compared to an estimated MPI of 0.091 for the South Asia region based on surveys between 2011 and 2022 (Alkire et al., 2023).

In the Nepali context, most previous research and building of databases by academics, government organisations, and non-governmental organisations (NGOs) have centred on the impacts of single hazards (Bhatta and Adhikari, 2024). There is an increasing shift from single hazard to multi-hazard approaches, exemplified by studies such as the multi-layer risk assessment of Khatakho et al. (2021) in the Kathmandu Valley that superimposed earthquake, fire, flood, and landslide

risk. In response to recent multi-hazard events such as the Melamchi debris flow in 2021, part of a hazard cascade that caused multi-sectoral impacts on a regional scale across 1 year (Sharma et al., 2023), more research is focused on multi-hazard impact and risk assessments (e.g. Dunant et al., 2024). Examples of the breadth of natural hazard events in Nepal and subsequent cascading hazards or impacts include the following:

- The high-profile disaster of the Gorkha earthquake in April 2015 caused devastating impacts in the Kathmandu Valley and beyond (Takai et al., 2016; Khatakho et al., 2021), with subsequent landslides on the periphery of the valley exacerbating these effects and prolonging the recovery effort (Kargel et al., 2015).

- Urban fires occur frequently and spread rapidly in areas of the Kathmandu Valley with high population density (Khatakho et al., 2021), particularly in informal settlements where marginalised communities experience disproportionate hazard impacts and may have lower capacity to prepare and respond to hazard events (Brown et al., 2019; Dodman et al., 2022).

- Both fluvial and pluvial flooding are frequent in the Kathmandu Valley during the monsoon season. For ex-

ample, during floods in early September 2021, heavy rainfall caused severe inundation across large areas of the valley and displaced hundreds of families in the Bansighat area (Chaulagain et al., 2023).

5 These earthquake, fire, flood, and landslide multi-hazard events exemplify the complexity of the interrelationships between hazards and their impacts and highlight the need to understand how these events relate to the geographical contexts in which they occur.

### 1.4 Multi-hazard data challenges in the Kathmandu Valley context

Similar to geographical contexts globally, across national income levels and the Global North and Global South categorisation, one of the significant challenges facing urban areas in 15 Nepal is the scarcity of disaggregated multi-hazard impact data, which is a barrier to effective DRR strategies (Panta, 2020; De Maio et al., 2024). Nepali hazard event databases, including the Nepal DRR Portal (2024) and BIPAD Portal (NDRRMA, 2024), predominantly document direct and tangible impacts, with some basic disaggregation by gender. For example, within the Nepal DRR Portal, one of Nepal's primary sources of damage and loss data, there are data gaps concerning spatial and temporal coverage, estimated losses, and incomplete loss indicators. These challenges are coupled with the restructuring of Nepal's administrative divisions in 2015, making spatial comparisons pre- and postrestructuring more difficult (Panta, 2020).

The Kathmandu Valley, Nepal, experiences a variety of single natural hazard and multi-hazard events against 30 the backdrop of significant urbanisation, rapid population growth, and climate-change-related challenges (Nehren et al., 2013; Pradhan-Salike and Pokharel, 2017). According to the Nepal National Population and Housing Census 2021, the Kathmandu Valley had a total population of 3 025 386 35 people, comprising 2 041 587 people in Kathmandu District (5169 people per square kilometre), 551 667 people in Lalitpur District (1433 people per square kilometre), and 432 132 people in Bhaktapur District (3631 people per square kilometre) (National Statistics Office, 2023). Within this popu-40 lation, marginalised communities, including residents of informal settlements under the broader grouping of urban poor communities, experience a disproportionate burden of hazard impacts due to their heightened socio-economic vulnerability (Pelling et al., 2004; Gorman-Murray et al., 2018; Dod-45 man et al., 2022). In mainstream hazard impact data, more vulnerable groups often lack representation (Osuteye et al., 2017). Addressing this data gap is crucial to prevent DRR strategies from unintentionally exacerbating existing social inequalities (Brown et al., 2019). To illustrate CE2 TS2 the spa-50 tial distribution of socially marginalised communities in one region of the Kathmandu Valley, Fig. 1 presents six informal settlements, with a total population of 7270 people in 2019 (Khanal and Khanal, 2022), out of 53 informal settlements

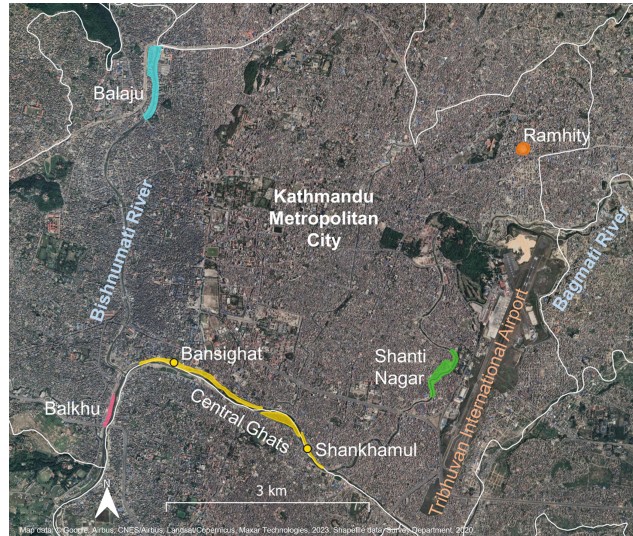

**Figure 1.** Map showing the location of six selected informal settlements in the Kathmandu Valley, Nepal (Balaju – solid blue; Shanti Nagar – solid green; Ramhity – solid orange; Balkhu – solid pink; Central Ghats – solid yellow); main rivers (Bishnumati and Bagmati); Tribhuvan International Airport (orange text); and administrative boundaries (white solid line). Locations of informal settlements are adapted from Dowse et al. (2014). Map data: © Google, Airbus, CNES/Airbus, Landsat/Copernicus, Maxar Technologies, 2023. Shapefile data: Survey Department, 2020.

that are documented in the valley (DUDBC, 2010). Within the Kathmandu Valley, 35 of 53 (66 %) informal settlements 55 are located along the banks of significant river corridors like the Bagmati (Fig. 1). The Kathmandu Valley experiences a breadth of single natural hazard types (Pradhan et al., 2020; Whitworth et al., 2020; Khatakho et al., 2021), with the potential for interrelationships to occur between these hazards 60 and across varying spatial and temporal scales.

In this paper, we systematically develop an overview of single hazard types and multi-hazard interrelationships influencing the Kathmandu Valley in our Kathmandu Valley Single Hazards and Multi-Hazard Interrelationships Database 65 (Thompson et al., 2024). These sources provide evidence of hazards that have already influenced the Kathmandu Valley and those that could potentially influence it, with descriptions of impacts where available in the sources. We supplement these blended source types with a workshop of practitioner 70 stakeholders engaged in DRR in the Kathmandu Valley. We used a similar methodology applied in Nairobi, Kenya, and Istanbul, Türkiye, in the context of the GCRF Tomorrow's Cities Hub (https://tomorrowscities.org/, last access: 2 December 2024), under which part of this research has been 75 conducted, which also looked at single hazard types and multi-hazard interrelationships in both hub cities (Šakić Trogrlić et al., 2024b). Subsequent sections are organised as follows: Sect. 2 develops the methodology, and Sect. 3 describes the results, followed by a discussion of findings in Sect. 4. We 80

suggest that our methodology can support the understanding of hazardscapes in other data-scarce urban contexts.

## 2 Methodology

This section outlines our methodology for creating a database of single hazard and multi-hazard interrelationship exemplars in a low-data-availability context. Our methodology focuses on natural hazards and does not include technological, environmental, or biological hazards as defined by the UNDRR (2017b), apart from urban fire owing to the high risk of incidence in the Kathmandu Valley. Our Kathmandu Valley Single Hazards and Multi-Hazard Interrelationships Database (Thompson et al., 2024) is based on blended sources comprising different evidence types. Although collating sources primarily aimed to evidence single hazard types and multi-hazard interrelationships influencing the Kathmandu Valley, we also reviewed all selected sources. We noted impacts when they were described (Column 9, Sheet "A. Single Hazards Evidence" and Column 6.1, Sheet "B. Hazard Interrelationships Evidence"). In the context of this paper, we note that "influence" refers to a single hazard or multi-hazard interrelationship occurring in or having a theoretical possibility of occurrence in the Kathmandu Valley, whereas "impact" refers to this occurrence or the theoretical possibility of occurrence realising consequences that affect the Kathmandu Valley. A workshop with practitioner stakeholders engaged in DRR in the context of Nepal supplements the database to add richness and incorporate additional multi-hazard interrelationship scenarios not included in the original exemplars.

Researchers are increasingly integrating blended (varied) sources of evidence to collate hazard events and their impacts. This use of blended sources of evidence is similar to Gill et al. (2020), who used a "comprehensive, systematic, and evidenced" approach to create a framework for multi-hazard interrelationships on a regional scale. Gustafsson et al. (2023) and Šakić Trogrlić et al. (2024b) adopted this methodology to compile natural hazard interrelationships for Sweden and for Nairobi and Istanbul, respectively.

Our study builds upon the methodology of Gill et al. (2020) and develops it further to include multi-hazard interrelationship data on a much finer spatial scale. In the case of the Kathmandu Valley, this equates to ward level and individual urban poor settlements as the highest spatial resolution of data collated from the blended sources. The issue of data availability becomes more complex at finer spatial scales (Osuteye et al. 2017), which our study contributes towards resolving.

Section 2.1 outlines the commonalities between searches undertaken for single hazard types and multi-hazard interrelationships. Section 2.2 describes specific information relevant to the searches for single hazard types and Sect. 2.3 the multi-hazard interrelationships. Section 2.4 closes with a description of a workshop undertaken with practitioner stakeholders engaged in DRR in the Kathmandu Valley.

### 2.1 Systematic mapping of single hazard types and multi-hazard interrelationships

This work builds upon previous studies utilising blended evidence sources to systematically review and collate multi-hazard events and impacts and supplement these sources with practitioner stakeholder knowledge (Neri et al., 2008; Gill et al., 2020; Taylor et al., 2020). Our methodology aims to be systematic but not comprehensive in gathering evidence for the potential for a specific single hazard or multi-hazard interrelationship relevant to the Kathmandu Valley. Our methodology followed elements of a systematic mapping process, which aims to systematically find, evaluate, and integrate evidence using predefined guidelines (Grant and Booth, 2009; James et al., 2016). We outline the methodological steps as follows:

- We searched for exemplars of single hazard types and multi-hazard interrelationships that either have influenced or could influence the Kathmandu Valley in academic literature, grey literature (e.g. national society, NGO/INGO, and multilateral development bank reports), media, databases, and social media (e.g. YouTube videos).

  - Here, we define grey literature as material produced outside of commercial publishers: "that which is produced on all levels of governmental, academics, business and industry in print and electronic formats, but which is not controlled by commercial publishers" (Auger, 1998).

  - We did not specify the spatial boundary around the Kathmandu Valley; instead, we considered case studies relevant if they directly or indirectly influenced people, the economy, infrastructure, or the environment in the valley.

- Variations of the single hazard terms (e.g. singular and plural) and Kathmandu were used to find evidence for the single hazard types (e.g. "landslide* AND Kathmandu", where * represents zero or more characters (e.g. landslide, landslides)).

- A simple Boolean search was used to find evidence for the multi-hazard interrelationships, with the following keywords: the multi-hazard interrelationship type AND "Kathmandu" AND impact*.

  - An example of an earthquake triggering or increasing the probability of a landslide would be "earthquake* AND landslide* AND Kathmandu AND impact*".

- We conducted searches in English as the authors do not speak Nepali, although, ideally, we would complete the search in both languages.

– We applied our literature searches to two online databases of academic publications (Web of Science and Google Scholar). If these returned no results, we conducted similar searches in three online English language Nepali newspapers (*The Himalayan Times*, *The Kathmandu Post*, and *Nepali Times*), global and national disaster databases (e.g. EM-DAT, Nepal DRR Portal), and YouTube using their built-in search tools.

- We reviewed the literature for each source type (academic literature, grey literature, media, databases, and social media (e.g. YouTube videos)), and where available, we chose publications from 2010 onwards. We selected only one publication as exemplar evidence for each single hazard type (Sheet "A. Single Hazards Evidence") and one publication as exemplar evidence for each multi-hazard interrelationship type (Sheet "B. Hazard Interrelationships Evidence") in the Kathmandu Valley Single Hazards and Multi-Hazard Interrelationships Database (Thompson et al., 2024).

- We focused on more recent events to capture the present hazardscape and its evolution. However, if recent events were unavailable, we searched for information on hazard events extending further back in time.

- If the Boolean search returned no results with this iteration, we omitted "impact*" from the search string and conducted the searches again in the same search engines.

– We then selected exemplars of hazard events from these blended sources of academic literature, grey literature, media, databases, and social media.

- When searches for a specific single hazard type or multi-hazard interrelationship returned more than 10 results per search engine, we skimmed titles and abstracts for an indication of spatial and temporal occurrence, and we selected up to five pieces of evidence that documented previous influence in the Kathmandu Valley (case studies) or had a theoretical possibility of influencing the Kathmandu Valley.

- If this examination of case study evidence returned no results for a particular hazard, we searched for any indication that the hazard may be theoretically possible of imparting impact in or around the Kathmandu Valley.

- Where we found no examples of specific single hazards or multi-hazard interrelationships impacting the Kathmandu Valley, such as "tornado*" or

**Table 2.** Classification of the 6 hazard groups and 23 single natural hazards. Adapted from the original classification developed by Gill and Malamud (2014).

| Hazard group | Hazard | Code |
| --- | --- | --- |
| Geophysical hazards | Earthquake | EQ |
| | Tsunami | TS |
| | Volcanic eruption | VO |
| | Landslide | LA |
| | Snow avalanche | AV |
| Hydrological hazards | Flood | FL |
| | Drought | DR |
| Shallow Earth processes | Regional subsidence | RS |
| | Ground collapse | GC |
| | Soil subsidence | SS |
| | Ground heave | GH |
| Atmospheric hazards | Storm | ST |
| | Fog | FO |
| | Tornado | TO |
| | Hailstorm | HA |
| | Snowstorm | SN |
| | Lightning | LN |
| | Extreme temperature (heat) | ET(H) |
| | Extreme temperature (cold) | ET(C) |
| Biophysical hazards | Wildfire | WF |
| | Urban fire | UF |
| Space/celestial hazards | Geomagnetic storm | GS |
| | Impact event | IM |

"geomagnetic storm*", we searched for evidence of the hazard occurring within or having a theoretical possibility of occurrence in Nepal with recorded or potential impacts in the Kathmandu Valley.

## 2.2 Single hazard types

To decide which single hazard types we would apply our systematic searches (Sect. 2.1) of blended evidence to, we started with the categorisation that Gill and Malamud (2014) developed with 6 hazard groups and 21 single natural hazards. There are other classifications, such as the hazard types in the UNDRR–ISC hazard information profiles (Murray et al., 2021). We added fog to the atmospheric hazard group and urban fire to the biophysical hazard group – where the nature of their fuel distinguishes urban fire and wildfire. Our hazard categorisation for the Kathmandu Valley comprised 6 hazard groups divided into 23 single natural hazards (Table 2).

The six hazard groups used here (Table 2) were geophysical, hydrological, shallow Earth processes, atmospheric, biophysical, and space/celestial hazards. We used the methodology described in the literature review in Sect. 2.1 to conduct Boolean searches for potential single hazard types and multi-hazard interrelationships that have influenced or could potentially influence the Kathmandu Valley. The examples

of single hazard types (Sheet "A. Single Hazards Evidence") were collated into the Kathmandu Valley Single Hazards and Multi-Hazard Interrelationships Database (Thompson et al., 2024) to summarise the evidence collected. A subset of this database is shown in Fig. 2. Each Excel row in Fig. 2 details evidence of a single hazard influencing the Kathmandu Valley. Column thematic groups in the database include hazard type, source information and link, source content, multi-hazard interrelationships and anthropogenic processes, video evidence, source and major event typical frequency reflections, and impact.

### 2.3 Multi-hazard interrelationships

Applying a similar systematic search to that described in Sect. 2.2 above, we collated exemplars of multi-hazard interrelationships that have either influenced or could potentially influence the Kathmandu Valley. Following the visualisation matrix developed by Gill and Malamud (2014), the 23 single hazard types included in the methodology could interact to produce a maximum number of $23 \times 23 = 529$ theoretically possible interrelationships. Some of the single hazard types or their interrelationships do not apply to the Kathmandu Valley (e.g. tsunami and snow avalanche) or have a low probability of occurrence (e.g. impact event triggering earthquake, volcanic eruption triggering or increasing the likelihood of earthquake). We documented which low-probability events have a theoretical chance of occurring in or could influence the Kathmandu Valley. These interrelationships may be omitted from government or community preparedness plans yet pose significant impacts if they occur, especially if strategies are not in place to mitigate the effects.

Cognisant of these single hazard types, we searched the literature to determine how many theoretically possible multi-hazard interrelationships have evidence of influence in the Kathmandu Valley. We focused on two types of multi-hazard interrelationships (Gill and Malamud, 2017, p. 261):

- *Triggering relationship*. "One primary natural hazard triggers a secondary natural hazard."

- *Increased-probability relationship*. "One primary natural hazard increases the likelihood of a secondary natural hazard."

We chose to search for triggering and increased-probability multi-hazard interrelationships (consecutive hazards) to build on the same methodology used by Gill et al. (2020) and to increase the number of returned search results compared to less well-documented multi-hazard interrelationship types, such as compound or coincident hazards. We collated the multi-hazard interrelationship results (Sheet "B. Hazard Interrelationships Evidence") in the Kathmandu Valley Single Hazards and Multi-Hazard Interrelationships Database (Thompson et al., 2024) with primary and secondary hazard rows – an extract of the database is given in Fig. 3.

This database includes detailed source information to gauge the reliability of the sources used to populate the database. Columns in the database are listed in the figure caption.

### 2.4 Workshop on multi-hazard interrelationships and impacts

We facilitated a workshop to supplement the single hazard types and multi-hazard interrelationships collated using the blended source types and examine their impacts. This 2 h workshop, "Multi-hazard Interrelationships and Impacts in Kathmandu Valley", took place on 12 April 2023 with seven participants engaged in DRR in the Kathmandu Valley (King's College London Research Ethics, registration no. MRSP-21/22-26736). The workshop was organised as follows:

- a presentation on single hazard types and multi-hazard interrelationships in the context of the Kathmandu Valley (30 min)

- activities to gather participant perspectives on the Kathmandu Valley hazardscape (40 min)
  - multi-hazard scenario activities:
    a. group discussion: multi-hazard scenario generation (10 min)
    b. individual input in Padlet (10 min)
    c. group discussion: broad themes and synthesis of multi-hazard scenarios (10 min)
  - multi-hazard impact activity:
    d. individual input in Padlet (10 min).

In the discussions, participants were encouraged to elaborate on details of the examples they shared, such as the magnitude and duration of the events, and to consider the nature of the interrelationships (i.e. triggering, increased-probability, and compound hazards). The discussions also aimed to examine the multi-hazard interrelationships and impacts that participants considered significant in the Kathmandu Valley context and to relate these to the broader hazardscape.

The virtual Teams workshop aimed to co-produce multi-hazard interrelationship scenarios and their impacts through two workshop activities. To minimise the potential effect of power asymmetries (Secor, 2010; Wolf, 2018), we balanced the number of Nepali or Nepal-based (four) and British or UK-based (three) participants and female (two) and male (five) participants to support participants in feeling comfortable to share their knowledge and perspectives. These participants were selected based on their in-depth knowledge of single hazards and multi-hazard interrelationships in the Kathmandu Valley context, as well as existing connections built on pre-established working relationships (Wilmsen, 2008). Participants were drawn from the NGO or international non-governmental organisation (INGO), national society, research institute, and academic sectors. The research

| 1. Hazard type | | | | 2. Source information and link | | | |
|---|---|---|---|---|---|---|---|
| 1.1 Hazard group | 1.2 Hazard | 1.3 Code | 1.4 Component hazards | 2.1 Case Study (C)/ Review (R)/ | 2.2 Type of evidence: Academic (A); Grey Lit (G); Media (M); Database (D); Social Media (SM); Not Applicable (NA) | 2.3 Source | 2.4 Link |
| Geophysical | A. Earthquake | EQ | Ground Shaking, Ground Rupture, Liquefaction, Co-Seismic Subsidence and Uplift | C | A | Takai, N., Shigefuji, M., Rajaure, S., Bijukchhen, S., Ichiyanagi, M., Dhital, M.R. and Sasatani, T. (2016) Strong ground motion in the Kathmandu Valley during the 2015 Gorkha, Nepal, earthquake. *Earth, Planets and Space*, 68(1), pp.1-8. | https://earth-planets-space.springeropen.com/articles/10.1186/s40623-016-0383-7 |
| | | | | C | SM | United Nations (2015) Nepal: The Gorkha Earthquake [Online] Available from: https://www.youtube.com/watch?v=gjzuNXN0R78 [Accessed 12 August 2021] | https://www.youtube.com/watch?v=gjzuNXN0R78 |

| 3. Source content | | 4. Hazard interrelationships and anthropogenic influences | | 5. Video evidence |
|---|---|---|---|---|
| 3.1 If case study, area impacted [Nepal, Kathmandu valley, Kathmandu] | 3.2 Description of the themes covered in the source | 4.1 Interrelationships with other hazards mentioned in the source (with addition of bolding in the quotes) | 4.2 Anthropogenic processes and influences mentioned in the source | 5.1. Illustrative YouTube video of case study or type of hazard discussed [Note YouTube videos can also be used as separate source] |
| Kathmandu | • Example of **ground shaking** and **ground rupture** in Kathmandu - **Mw 7.8 on 25 April 2015** • Epicentre occurred 80 km northwest of Kathmandu in Gorkha region • Seismic response of soft lake sediments is main factor for 'significant damage' | — | — | https://www.youtube.com/watch?v=gjzuNXN0R78 |
| Kathmandu | • Overview of the **Gorkha earthquake** in 2015 • Summary of major **impacts** and **short-term responses** in Kathmandu | — | • United Nations World Food Programme **distributed food** to over **300,000** people, particularly those in remote areas | NA |

| 6. Source reflections | | | 7. Major event typical frequency reflection | 8. Any other reflection on a single hazard | 9. Impact |
|---|---|---|---|---|---|
| 6.1. Any other comments on the source | 6.2 How much is the hazard evidenced by different types of sources (e.g., Earthquakes impacting Kathmandu primarily mentioned in peer-review literature) | 6.3 Difficulty in finding sources 1 = Frequent Kathmandu specific sources across multiple types of evidence 2 = Frequent Kathmandu specific sources, limited to a few main types of evidence 3 = Kathmandu specific sources are scarce 4 = Sources are scarce, focused on regional scale hazards 5 = Sources are very scarce, focused on national scale or very rare hazards | 7.1. Reflection on how frequently a hazard of a given size occurs (e.g.earthquakes of a magnitude X or larger occur about every Y years) (Text in blue letters is input from the __) | 8.1. Additional reflection by local stakeholders (Text in blue letters is input from the __) | 9.1 Impact on the documented area |
| Search term: 'Kathmandu earthquake' in Google Scholar (GS) | • Frequently mentioned in the academic literature, media and databases - predominantly Gorkha 2015 earthquake. | 1 | • Medieval earthquake magnitudes are unknown, Mw ~7.8 in 1833, Mw 8.2 in 1934 and Mw 7.8 in 2015 • Earthquake clusters from 12-14th centuries and 19th century onwards • 1833 and 1934 earthquakes caused liquefaction 'on a large scale', 2015 earthquake caused moderate liquefaction in the Kathmandu Valley versus minimal liquefaction in the distal Bihar Plains (Rajendran, 2021) | | • **8000 fatalities**, majority of which were in Kathmandu |
| Search term: 'Gorkha earthquake' in YouTube | | | | | • More than **8000 fatalities, 17,000 people injured** • **75% of buildings** in Kathmandu **destroyed** • **Tens of thousands** of people made **homeless** - housed in **temporary shelters** • **2 million** children and their families needed **immediate help** - many **children separated** from their families |

**Figure 2.** Extract of the Kathmandu Valley single hazard types evidence from Sheet "A. Single Hazards Evidence" in the Kathmandu Valley Single Hazards and Multi-Hazard Interrelationships Database (Thompson et al., 2024) for a section of the geophysical hazard group. Shown are the two header rows and two of the rows of single hazard exemplar information (for earthquakes) out of 60 rows.This includes information on (1.1) hazard group, (1.2) hazard type, (1.3) shorthand code of the hazard, (1.4) component hazards, (2.1) case study/review, (2.2) type of evidence, (2.3) source, (2.4) link to the source, (3.1) if case study, area influenced, (3.2) description of the themes covered in the source, (4.1) interrelationships with other hazards mentioned in the source, (4.2) anthropogenic processes, (5.1) YouTube evidence, (6.1) additional comments on the source, (6.2) how much the hazard is evidenced by different types of sources, (6.3) difficulty in finding sources, (7.1) major event typical frequency reflection, (8.1) additional reflection by local practitioner stakeholders, and (9.1) impact on the documented area.

backgrounds of participants were social scientists, physical scientists (e.g. geography) CE3, and interdisciplinary scientists (e.g. thematic lead: climate and resilience) and ranged from early career (e.g. research associate) to senior career (e.g. professor). We utilised the snowball sampling technique (Secor, 2010) to encourage participants to suggest any further colleagues who they thought might be interested in participating in the same workshop for us to contact. All participants gave informed consent for participation, indicating

| 1. Hazard type | | | | 2. Source information and link | | |
|---|---|---|---|---|---|---|
| **1.1 Primary hazard** | **1.2 Secondary hazard** | **1.3 Grid ID** | **1.4 Generic mechanism description** | **2.1 Example from Kathmandu Valley** | **2.2 Link to source** | **2.3 Source type Academic (A); Grey Lit (G); Media (M); Database (D); Social Media (SM); Not Applicable (NA)** |
| **Earthquake** | Earthquake | 1A | A primary earthquake causes changes in lithospheric stresses, leading to aftershocks as the lithosphere responds to these changes. | Lizundia, B., Davidson, R.A., Hashash, Y.M. and Olshansky, R. (2017) Overview of the 2015 Gorkha, Nepal, earthquake and the earthquake spectra special issue. *Earthquake Spectra*, 33(1_suppl), 1-20. | https://journals.sagepub.com/doi/pdf/10.1193/120817eqs252m | A |
| | | | | Rafferty, J. P. (2021) Nepal earthquake of 2015. *Encyclopedia Britannica*. [Online] Available from: https://www.britannica.com/topic/Nepal-earthquake-of-2015 [Accessed 1 September 2021]. | https://www.britannica.com/topic/Nepal-earthquake-of-2015 | G |

| 3. Source content | | | | 4. Hazard sequence | | 5. Source reflections | | |
|---|---|---|---|---|---|---|---|---|
| **3.1 Interrelationship type: Triggered (T)/ Increased probability (I)/ Both (B)/Other (O)** | **3.2 Case study (C)/ Theoretically possible in Kathmandu Valley (P)** | **3.3 Description** | **3.4 Any additional comments (e.g. more than one hazard, future developments and planning)** | **4.1 Hazard sequence** | **5.1 Search criteria** | **5.2 How much is the interrelationship evidenced by different types of sources? (e.g., earthquakes triggering landslides primarily mentioned in peer-review literature).** | **5.3 Difficulty in finding sources 1 = Frequent Kathmandu specific sources across multiple types of evidence 2 = Frequent Kathmandu specific sources, limited to a few main types of evidence 3 = Kathmandu specific sources are uncommon 4 = Sources are scarce, focused on regional scale hazards 5 = Sources are very scarce, focused on national scale or very rare hazards** | |
| T | C | • Mw 7.8 earthquake with epicentre near Gorkha, with **4 aftershocks >Mw 6.0** occuring by March 2016. • Largest magnitude aftershock was Mw 7.3 on 12 May 2015 recorded 140 km southeast of the main earthquake. • Damage occurred in Kathmandu. | • Earthquake triggered aftershocks AND liquefaction AND landslides/ground failure. | • earthquake -> landslides • earthquake -> aftershocks -> landslides • earthquake -> liquefaction | Kathmandu AND earthquake* AND aftershock* AND impact* (Web of Science) | • Earthquakes triggering earthquakes are found across evidence types, although there is a strong focus on the 2015 Gorkha earthquake, secondary hazards and impacts. | 1 | |
| T | C | • Details the Gorkha earthquake and **aftershocks** in **2015**, and their impacts in Kathmandu. • Within one day of the main earthquake, two **aftershocks** of magnitudes **6.6** and **6.7** occurred, followed by "dozens" of smaller magnitude aftershocks over the following days. | • The initial earthquake triggered **landslides** that devastated "some of the **most densely populated parts of Kathmandu**. | • earthquake -> earthquake • earthquake -> landslide | Kathmandu AND earthquake* AND earthquake* (Ecosia) | | | |

| 6. Impact | 7. Input from practitioner stakeholders |
|---|---|
| **6.1 Impact** | **7.1 Input from practitioner stakeholders 1. Is the identified interrelationship relevant for/applicable to Kathmandu Valley? 2. Would you classify the identified interrelationship as important for today's Kathmandu Valley? 3. Is this interrelationships relevant for future Kathmandu Valley (e.g., will become of increasing importance) and should be taken into account in urban planning?** |
| • Over **8,790 fatalities** and **22,300 injuries**. • ~**Half a million homes** destroyed. • Hundreds of **historical and cultural sites** destroyed/severely damaged. • **Infrastructure damaged**, such as roads and hospitals. | |
| • Approximately **9000 fatalities**, **16,800 injuries** and **2.8 million people** displaced. • The UN reported that over **8 million people** (or 1/4 of Nepal's population) were **"affected by the event and its aftermath"**. • Initial damage estimates were **$5 billion to $10 billion**. | |

**8.1 Input from practitioner stakeholders - prioritisation**

Based on your inputs in Column Q, can you please:

**1) Indicate the most important hazard interrelationships in today's Kathmandu Valley:**

-
-
-

**2) List hazard interrelationships which you feel will be relevant for tomorrow's Kathmandu Valley (including the interrelationships which are already relevant today):**

-
-
-

**Figure 3.** Extract of the Kathmandu Valley multi-hazard interrelationships evidence from Sheet "B. Hazard Interrelationships Evidence" in the Kathmandu Valley Single Hazards and Multi-Hazard Interrelationships Database (Thompson et al., 2024) for the earthquake (primary hazard) to earthquake (secondary hazard) section. Shown are the two header rows and two of the rows of the multi-hazard interrelationship exemplar information out of 95 rows. This includes information on (1.1) primary hazard, (1.2) secondary hazard, (1.3) grid ID, (1.4) generic mechanism description, (2.1) example from the Kathmandu Valley, (2.2) link to source, (2.3) source type, (3.1) interrelationship type, (3.2) case study/theoretically possible in the Kathmandu Valley, (3.3) description, (3.4) any additional comments, (4.1) hazard sequence, (5.1) search criteria, (5.2) how much the multi-hazard interrelationship is evidenced by different source types, (5.3) difficulty in finding sources, (6.1) impact, (7.1) input from practitioner stakeholders, and (8.1) input from practitioner stakeholders – prioritisation.

their requested level of anonymity (i.e. any combination or none of the following: full name, position, and institution).

During the workshop, we prompted discussion and knowledge sharing on multi-hazard interrelationships and multi-hazard impacts that could influence the Kathmandu Valley in the future. We investigated these themes through group discussion in the main Teams meeting room, supported by the chat function, and individual input into Padlet (Fig. 4). The virtual pinboard within Padlet (Fig. 4) is an online resource that supports sharing ideas by posting content on a shared web page. In the Padlet, we prompted participants to share their perspectives on the following themes:

1. Multi-hazard interrelationship scenarios

   – Examples of case studies and theoretical multi-hazard scenarios (triggering, increased-probability, and compound hazards) in the Kathmandu Valley;

   – Additional information about multi-hazard scenarios (e.g. magnitude, duration).

2. Multi-hazard impacts

   – Examples of multi-hazard impacts in the Kathmandu Valley and their significance;

   – Identify which social groups are most vulnerable to these impacts.

Verbal group discussions were semi-structured to support participants in sharing their perspectives on the themes with prompts to guide the conversation where needed. As facilitators, we aimed to balance the contribution of each participant to minimise domination of the discussions by one or more participants and to ensure that all participants felt comfortable sharing their thoughts. Many researchers have emphasised that when we work with individuals and communities, we must be aware of the contradictions of conducting fieldwork that centres equity and social justice in contexts where structural power imbalances exist between the researcher and those individuals and communities on whom the research is focused (Subedi, 2006; Ozkazanc-Pan, 2012; Manning, 2018; Wolf, 2018). Throughout this study, we were reflexive about our positionalities, how our "otherness" contributes towards power relations (Subedi, 2006; Mishra, 2018), and how we could minimise these effects in our research. Further discussion of positionality is presented in Sect. 4.4.2 and explores the impact of who was present or not in the room on the workshop results.

## 3 Results

Here, we give the results of single hazard (Sect. 3.1) and multi-hazard interrelationship exemplars (Sect. 3.2) that have or could influence the Kathmandu Valley as documented in blended source types. This is followed by insights from workshop participants (Sect. 3.3), including multi-hazard scenarios and their impacts.

### 3.1 Single hazard types influencing the Kathmandu Valley

Using the methodology described in Sect. 2.1 and 2.2, we compiled 58 sources of evidence for single hazard types (Sheet "A. Single Hazards Evidence") in the Kathmandu Valley Single Hazards and Multi-Hazard Interrelationships Database (Thompson et al., 2024) that have or might influence the Kathmandu Valley. These sources evidenced 21 of the 23 single hazard types given in Table 2, not including the following:

– *Tsunamis.* There are no large lakes or bodies of water near enough or in the Kathmandu Valley for tsunami occurrence.

– *Snow avalanches.* The incidence of heavy snow is very infrequent in the Kathmandu Valley. In the Nepal DRR Portal, there are no recorded snow avalanche events in Bhaktapur, Kathmandu, and Lalitpur (the three districts comprising the Kathmandu Valley) from DesInventar Sendai (UNDRR, 2024) or the Ministry of Home Affairs (Nepal DRR Portal, 2024).

For the 21 single hazard types evidenced that might influence the Kathmandu Valley, academic literature comprised the largest proportion of source types across the single hazards. All the sources were published from 2010 to 2021; the only exception was an example of soil subsidence published in 2002, as evidence of this hazard was challenging to find. Additionally, there have been no recorded tornadoes in the Kathmandu Valley to date. However, we included the tornado hazard type in the database due to the occurrence of windstorm Parvana in southeastern Nepal in March 2019. Researchers from The Small Earth Nepal and Nepal's Department of Hydrology and Meteorology reported the windstorm as Nepal's first recorded tornado (Mallapaty, 2019; Gautam et al., 2020).

Across the single hazard types, those that occur most frequently (e.g. flood, urban fire) or recently occurred with higher magnitudes (e.g. earthquake) were the most common hazard types reported by different sources, and the impacts of the hazards were often described in more detail. Conversely, it was more challenging to find evidence for hazard types where (a) major events occur less frequently (or have no direct evidence of occurrence; i.e. may only be theoretically possible in the Kathmandu Valley), such as volcanic eruptions or impact events, or (b) where typical "major" events have localised impacts and are considered an everyday occurrence by the local population, such as soil subsidence. These limitations are explored further in Sect. 4.4.

As illustrated by the results for single hazard types, the Kathmandu Valley is exposed to a plethora of hazard types,

notably earthquakes, urban fires, floods, and landslides (Gautam et al., 2021; Khatakho et al., 2021), owing to factors such as its tectonic location, high building density, position in the wider Bagmati river basin, incidence of the annual monsoon, and steep topography. In extracting major event typical frequency information from our analysis, shown in Table 3, we aim to give a preliminary indication of the prevalence and size of specific hazard types where this information is available, especially for hazard types that may be overlooked as a risk to the Kathmandu Valley.

Taking the typical frequencies of major events presented in Table 3 as exemplars rather than as an exhaustive compilation, these records suggest that specific hazards within the geophysical, hydrological, and atmospheric hazard groups have the most quantitative information on typical frequencies of major events in the context of the Kathmandu Valley. The remaining hazard groups – shallow Earth processes, biophysical, and space/celestial – have some information on typical frequencies of major events for some hazard types influencing the Kathmandu Valley, but this is typically limited to qualitative descriptions.

## 3.2 Multi-hazard interrelationships influencing the Kathmandu Valley

Searching for evidence of multi-hazard interrelationships that might influence the Kathmandu Valley focused on triggering and increased-probability relationships for primary to secondary hazards, using the 21 single hazard types in Sect. 3.1 as our primary and secondary hazards. Using the methodology given in Sect. 2.3, we found 83 multi-hazard interrelationships (out of a possible $21 \times 21 = 441$), of which 12 were directly evidenced using 21 blended sources (see Sheet "B. Hazard Interrelationships Evidence") in the Kathmandu Valley Single Hazards and Multi-Hazard Interrelationships Database (Thompson et al., 2024). We use a 23-cell × 23-cell (for completeness, we include tsunami and snow avalanche, which do not influence the valley) multi-hazard interrelationship matrix (Fig. 5) to visualise observed and theoretically possible interrelationships of primary to secondary hazards influencing the Kathmandu Valley:

– triggering only in 14 (17 %) of 83

– increased-probability only in 23 (28 %) of 83

– triggering and increased-probability in 46 (55 %) of 83.

This interrelationship matrix, definitions of hazards (Gill and Malamud, 2014), multi-hazard interrelationships (Gill and Malamud, 2014), and sources (Gill et al., 2020) are included in the Kathmandu Valley Single Hazards and Multi-Hazard Interrelationships Database (Thompson et al., 2024).

The multi-hazard interrelationship matrix in Fig. 5 visualises which scenarios have influenced or could potentially influence the Kathmandu Valley. We have justified interrelationships where a primary hazard increases the probability of urban fire (e.g. A21: earthquake increases the probability of urban fire) but not wildfire by considering anthropogenic processes as an intermediary step. For example, an earthquake could rupture gas mains or cause electricity pylons to fall, increasing the probability of an urban fire. Figure 5 provides an efficient tool for quickly assessing which multi-hazard interrelationship pairs or cascades of more than two hazards are relevant to the Kathmandu Valley context. Within these interrelationship types, the reader can rapidly determine the proportion of triggering interrelationships, increased-probability interrelationships, and both of these interrelationships and where we found direct evidence for a multi-hazard interrelationship influencing the Kathmandu Valley. In Fig. 6, we illustrate scenarios for two multi-hazard cascades (more than two hazards) that either have influenced or could potentially influence the Kathmandu Valley using the multi-hazard interrelationship matrix in Fig. 5.

The two multi-hazard scenarios in Fig. 6 clarify how to interpret which multi-hazard interrelationships are relevant to the Kathmandu Valley context, as shown in Fig. 5. For instance, the two multi-hazard scenarios show the interrelationships between single hazard types and hazard groups, where scenario (a) describes the cascade from a primary atmospheric hazard to a secondary geophysical hazard to a tertiary hydrological hazard and scenario (b) from a primary atmospheric hazard to a secondary hydrological hazard to a tertiary biophysical hazard. These two examples emphasise the interconnections between various Earth systems and the complex nature of interrelationships between single hazards, further supporting the need for holistic multi-hazard approaches to mitigating disaster risk. The dynamic multi-hazard scenarios (causal diagrams) given on the right-hand side of Fig. 6 are one of many scenarios that can be derived from the matrix. Two additional examples include the following:

– *Storm* triggers and increases the probability of *flood*.

– *Extreme temperature (heat)* increases the probability of *urban fire*.

The accompanying Kathmandu Valley Single Hazards and Multi-Hazard Interrelationships Database (Thompson et al., 2024) provides further information for each multi-hazard interrelationship (Sheet "B. Hazard Interrelationships Evidence") given in Figs. 5 and 6, including hazard type, source information, source content, hazard sequence, source reflections, impact, and input from practitioner stakeholders. This additional information provides greater context to the figures and enables the methodology to be scalable to other geographical regions, as discussed in Sect. 4.5.

## 3.3 Insights from workshop participants

We planned and facilitated a 2 h workshop with practitioner stakeholders engaged in DRR work in the Kathmandu Valley following the methodology described in Sect. 2.4. We

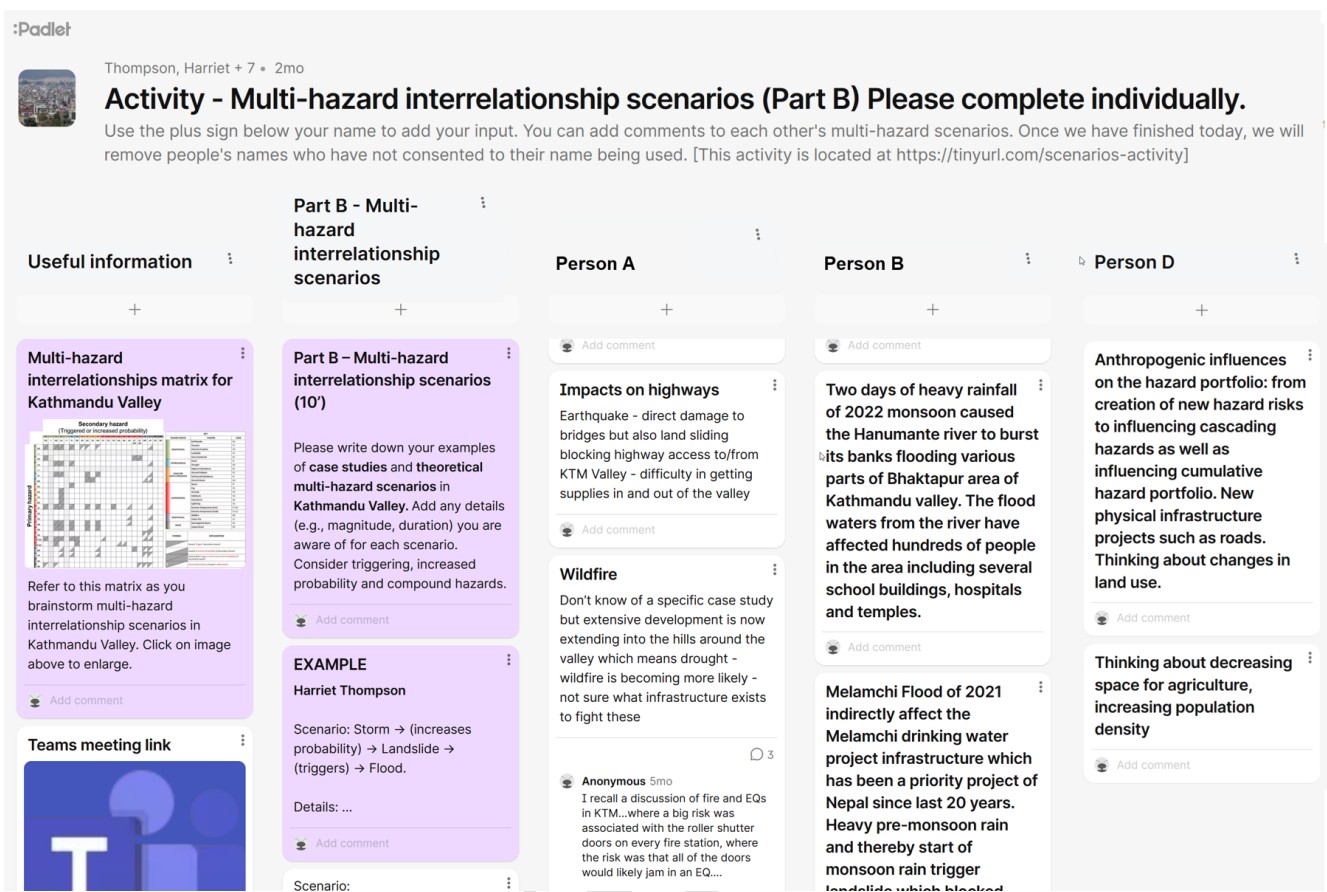

**Figure 4.** Portion of the populated virtual pinboard (Padlet) page used in the "Multi-hazard interrelationship scenarios" section of the 12 April 2023 workshop. The top of the image shows the title and brief description of the activity, with the participants' comments displayed below. We have replaced actual participant names with Person A–G (seven participants).

**Table 3.** Summary of major event typical frequencies for four single hazard types that could influence the Kathmandu Valley or Nepal.

| Hazard | Major events, including date and magnitude | Typical frequency of major events |
|---|---|---|
| Earthquake | 1255 (magnitude unknown), 1344 (magnitude unknown), 1833 ($M_w \sim 7.7$), 1934 ($M_w$ 8.2), and April 2015 ($M_w$ 7.8) earthquakes either influenced or likely influenced the Kathmandu Valley (Rajendran, 2021). | At least one major earthquake each century in Nepal (Tiwari and Paudyal, 2024). Magnitude $M_w$ 5.0–6.5 earthquakes in Nepal have a 5–10-year mean return period (Sharma and Biswas, 2024). |
| Volcanic eruption | 20 volcanic eruptions with VEI (volcanic explosivity index) = 6 to 8 were dated between 1.2 Ma to 1991 CE in Southeast Asia (De Maisonneuve and Bergal-Kuvikas, 2020). | Probabilities of VEI = 6, VEI = 7, and VEI = 8 volcanic eruptions occurring somewhere in Southeast Asia in 10 years are approximately 15 %, 1.2 %, and 0.1 % respectively (Whelley et al., 2015). |
| Flood | "Frequent" flooding in the Kathmandu Valley during the annual monsoon season (magnitudes not mentioned) (e.g. Chaudhary et al., 2024; Danegulu et al., 2024). | Annual daily maximum flood for 5-, 10-, and 25-year return periods in the Bagmati river basin, including the Kathmandu Valley, were estimated as 876, 1077, and 1331 $m^3\,s^{-1}$ under present climatic conditions (Mishra et al., 2024). |
| Tornado | Windstorm Parvana was the first recorded tornado in Nepal (mean speed: 250 $km\,h^{-1}$; estimated size: 200 $km^2$) (Chhetri et al., 2019). | Windstorm Parvana was the largest-scale storm in over 70 years in Nepal (Gautam et al., 2020). |

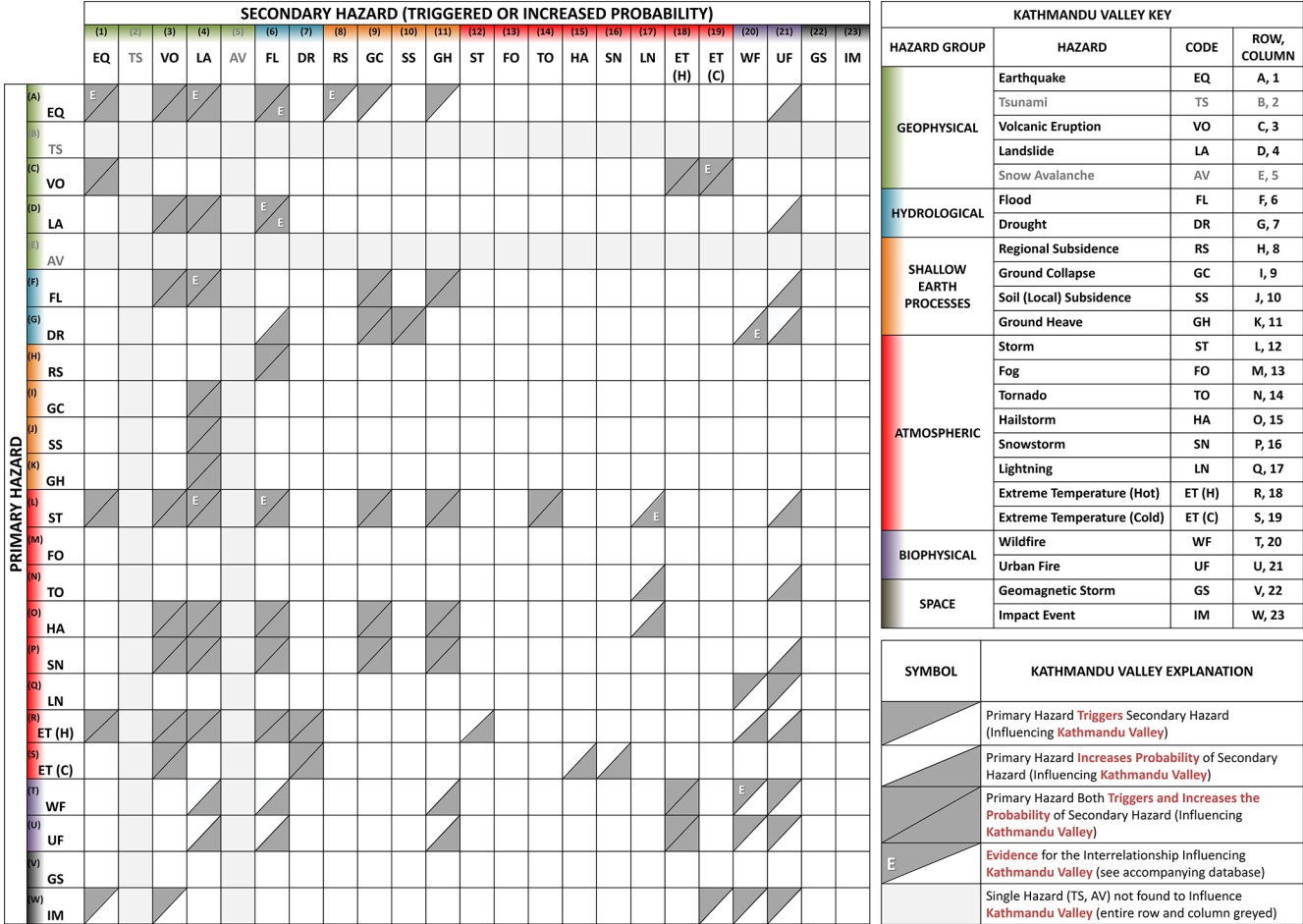

**Figure 5.** The 23-cell × 23-cell matrix of multi-hazard interrelationships that are theoretically possible in the Kathmandu Valley, Nepal, from the Kathmandu Valley Single Hazards and Multi-Hazard Interrelationships Database (Thompson et al., 2024). Primary hazards are on the $y$ axis, secondary hazards are on the $x$ axis, and the hazards are coded as detailed in the legend on the right. The hazards are categorised into geophysical (green), hydrological (blue), shallow earth processes (orange), atmospheric (red), biophysical (purple), and space (grey) hazard groups. The matrix shows where a primary hazard triggers a secondary hazard (upper left triangle shaded), a primary hazard increases the probability of a secondary hazard (lower right triangle shaded), a primary hazard both triggers and increases the probability of a secondary hazard (both triangles shaded), and where evidence is found for the interrelationship influencing the Kathmandu Valley (white letter E). This figure follows the visualisation and classification methodology developed by Gill and Malamud (2014) except that (i) tsunami and snow avalanche hazards are not found in the Kathmandu Valley and therefore not considered here for multi-hazard interrelationships (rows B and E and columns 2 and 5, greyed out), and (ii) fog and urban fire hazards are added as they are relevant in the Kathmandu Valley (rows M and U and columns 13 and 21).

supplemented the single hazard types and multi-hazard interrelationship blended evidence sources in the Kathmandu Valley Single Hazards and Multi-Hazard Interrelationships Database (Thompson et al., 2024) as follows. We designed the co-production of multi-hazard interrelationship scenarios to gather practitioner stakeholder perspectives on current applications of multi-hazard knowledge, opportunities for practitioner stakeholders to use multi-hazard scenarios, and implementation of these scenarios in DRR strategies in the Kathmandu Valley. We shared Fig. 6 with workshop participants to illustrate the value of the multi-hazard interrelationship matrix in extracting relevant multi-hazard scenar-

ios. Figure 6 was then made available virtually to participants during workshop discussions and acted as a visualisation tool and talking point to explore further multi-hazard interrelationships that have influenced or could influence the Kathmandu Valley beyond those archetypal examples that participants may associate with the region.

We present below the main results of the workshop: the production of multi-hazard interrelationship scenarios (Sect. 3.3.1) and impacts (Sect. 3.3.2). Participants discussed these contributions as a group before independently noting their examples on the Padlet pages. Through the workshop, we identified which multi-hazard interrelationships and im-

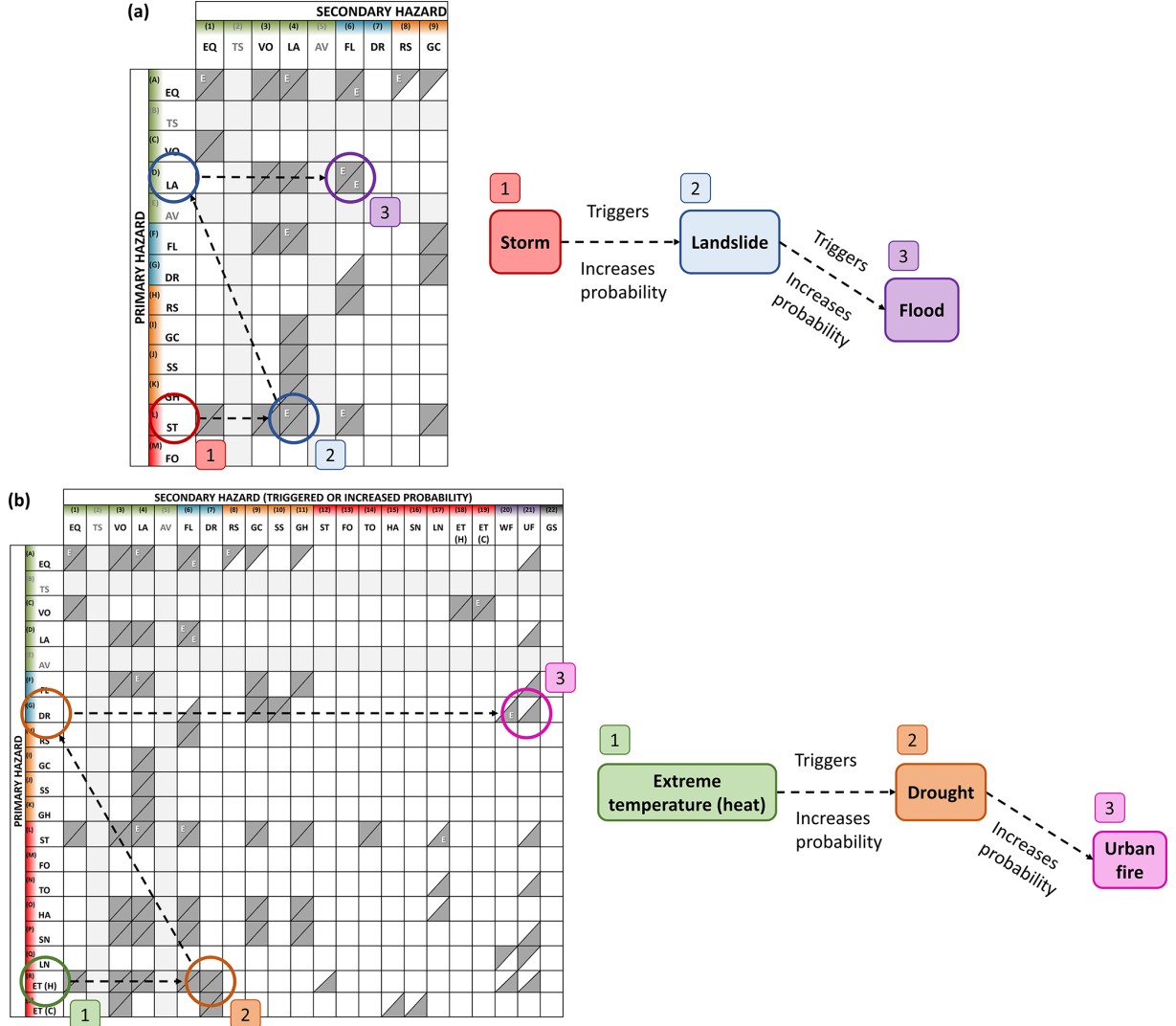

**Figure 6.** Two examples of scenarios for multi-hazard cascades that could influence the Kathmandu Valley, Nepal: **(a)** storm to landslide to flood and **(b)** extreme temperature (heat) to drought to urban fire. Both multi-hazard scenarios could involve triggering or increased-probability interrelationships and are illustrated using the multi-hazard interrelationship matrix (see Fig. 5). Multi-hazard scenario **(a)**, storm to landslide to flood, has evidence of influence in the Kathmandu Valley. Primary hazards are shown on the *y* axis, secondary hazards are displayed on the *x* axis, and the hazards are coded as detailed in the legend in Fig. 5. The matrix shows where a primary hazard triggers a secondary hazard (upper left triangle shaded), a primary hazard increases the probability of a secondary hazard (lower right triangle shaded), a primary hazard both triggers and increases the probability of a secondary hazard (both triangles shaded), and where evidence is found for the multi-hazard interrelationship influencing the Kathmandu Valley (white letter E). The multi-hazard interrelationship matrix is from the Kathmandu Valley Single Hazards and Multi-Hazard Interrelationships Database (Thompson et al., 2024).

pacts are most common, which are practitioner stakeholder priorities, and which are overlooked but may have significant implications in the future.

### 3.3.1 Multi-hazard interrelationship scenarios

5 The practitioner stakeholder workshop had an activity of 40 min where participants individually designed multi-hazard interrelationship scenarios and discussed these scenarios as a group. Participants individually added these

scenarios to a virtual pinboard (Padlet) page, also noting whether it was a case study that had previously influenced or a theoretical example that might influence the Kathmandu Valley in the future. Participants were encouraged to add any additional contextual information, such as comments about vulnerability and exposure, where known. Table 4 shows 11 multi-hazard interrelationship scenarios that the 7 participants shared on the workshop Padlet page.

Of the 11 multi-hazard interrelationship scenarios shared by participants (Table 4), the majority include hazards

**Table 4.** List of 11 multi-hazard interrelationship scenarios that include 2 or more hazards shared by participants on a virtual pinboard (Padlet) page during a practitioner stakeholder workshop. Each scenario is either a case study (CS) that has influenced the Kathmandu Valley or a theoretical example (TE) that could influence the Kathmandu Valley in the future. Any additional notes that provide context to the scenario are also given, including direct quotes and summaries of information shared by participants.

| Scenario number | Multi-hazard interrelationship scenario | Case study (CS) or theoretical example (TE) | Additional notes including participant quotes |
|---|---|---|---|
| 1 | Earthquake → damage to infrastructure + landslide → blocked supply access to the Kathmandu Valley | TE | |
| 2 | Drought → wildfire | TE | Exacerbated by development extending into the hills surrounding the valley. |
| 3 | Earthquake → urban fire | TE | Shutter doors in fire stations could jam in an earthquake and restrict fire engine access due to road debris. |
| 4 | Storm → flooding → infrastructure damage + transportation affected | CS | "Two days of heavy rainfall [during the] 2022 monsoon caused the Hanumante River to burst its banks flooding various parts of Bhaktapur area of Kathmandu Valley. The floodwaters from the river affected hundreds of people in the area including several school buildings, hospitals and temples." – Kathmandu Workshop Participant B. |
| 5 | Storm → landslide → flood | CS | "Melamchi Flood of 2021 indirectly affect[ed] the Melamchi drinking water project infrastructure which has been a priority project of Nepal [for the] last 20 years. Heavy pre-monsoon rain and thereby start of monsoon rain trigger[ed] landslide which blocked Melamchi river for about 45 min, and when that burst, flood [caused] by that had [a] significant impact on downstream communities." – Kathmandu Workshop Participant B. |
| 6 | Earthquake → (increasing probability of) → fire | TE | Consideration of the impact of seasonality on hazard cascades, here, an earthquake occurs in the dry season. |
| 7 | Earthquake → (cascading) → landslide → (cascading) → sedimentation → flood | TE | Consideration of the impact of seasonality on hazard cascades, here, an earthquake occurs in the monsoon season. |
| 8 | Storm → flood | CS | "Kalanki Settlement Area flooded in 2019. In 2019, heavy 3 d precipitation in Kathmandu resulted in flooding in the small stream near Kalanki city area. Water filled inside houses and roads were blocked. Many 2-wheeler vehicles swept away. Lots of the core city area in Kathmandu was flooded that time." – Kathmandu Workshop Participant F. |
| 9 | Wildfire → air pollution | CS | "Air pollution due to forest fire. In 2021, Nepal battled its worst forest fires in years. As per the officials, the fire smoke waft[ed] across mountains and sour[ed] the air as it settle[d] into the bowl that holds the capital city of Kathmandu. People were asked to stay in and Tribhuvan University class[es were] also suspended for few days to avoid pollution." – Kathmandu Workshop Participant F. |
| 10 | Haphazard development + faulty electrics → urban fire | TE | |
| 11 | Haphazard construction and development + soil conditions → ground displacement | TE | |

from the biophysical, geophysical, and hydrological hazard groups. Five scenarios include fire (biophysical), four include earthquake (geophysical), three include landslide (geophysical), and three include heavy rainfall or precipitation (hydrological). Although most (7 of 11) of the proposed scenarios are simple primary to secondary multi-hazard interrelationships, there are some examples of more complex interrelationships. Scenarios 6 and 7 in Table 4 illustrate how meteorological conditions can significantly influence the unfolding scenario and its impacts. If an earthquake occurs in the dry season, it is more likely to increase the probability of fire. Conversely, in the monsoon season, an earthquake could cascade into a landslide, increasing river sedimentation and the likelihood of flooding. These contrasts in environmental conditions illustrate how seasonality can significantly influence multi-hazard cascades.

In the 20 min group discussion, participants cited anthropogenic processes as influencing hazards. For example, Scenario 2 in Table 4 (drought → wildfire) states that development into the hills surrounding the Kathmandu Valley exacerbates the multi-hazard interrelationship and the resulting impacts. This challenge was echoed in Scenario 10 (haphazard development + faulty electrics → urban fire) and Scenario 11 (haphazard construction and development + soil conditions → ground displacement), which detail how rapid and sometimes "unmanaged" urbanisation increases the exposure of communities to multi-hazard events in the Kathmandu Valley.

The multi-hazard interrelationship scenarios proposed by participants support those in the multi-hazard interrelationship matrix (developed before the workshop) in Fig. 5. Scenario 4 (rainfall → flood) and Scenario 5 (rainfall → landslide → flood) in Table 4 are not included in the original matrix as rainfall is grouped under storm hazard. The same is true of Scenario 8 (rainfall → flood). The only three scenarios in Table 4 that are not present in the original matrix in Fig. 5 are those involving air pollution (Scenario 9: wildfire → air pollution) and processes of rapid urbanisation (Scenario 10: haphazard development + faulty electrics → urban fire, and Scenario 11: haphazard construction and development + soil conditions → ground displacement). Building upon Gill and Malamud's (2014) multi-hazard interrelationship framework, we focused on natural hazards in the multi-hazard interrelationship matrix – the only exception being urban fire due to the high risk of occurrence in the Kathmandu Valley. Including anthropogenic processes could form the basis of future developments of this work.

### 3.3.2 Multi-hazard impacts

In the final component of the 40 min workshop activity, we asked participants to individually add two or three examples of impacts from multi-hazard interrelationship scenarios in the Kathmandu Valley to a virtual pinboard (Padlet) page. We used the UNDRR (2017a) definition of (disaster) impact:

> The total effect, including negative effects (e.g. economic losses) and positive effects (e.g. economic gains), of a hazardous event or a disaster. The term includes economic, human and environmental impacts, and may include death, injuries, disease and other negative effects on human physical, mental and social well-being.

We requested that participants focus on impacts which they believed are most significant for people in the Kathmandu Valley and, where possible, to list which types of people (e.g. which social groups) might be most affected by these impacts. Table 5 shows 12 impact examples that the 7 participants shared on the Padlet page.

The 12 impact examples shared by workshop participants focus on hazard types that significantly impact people in the Kathmandu Valley: earthquakes, floods, landslides, and storms (monsoon rain and windstorms). Participants also described impacts that could more broadly apply to all hazard types that are theoretically possible in the Kathmandu Valley. Although some impacts shared by participants were direct and tangible, many examples considered indirect and intangible impacts, with attention focused towards the complex interrelationships between impacts and socio-political and anthropogenic processes. Indeed, the impact examples shared by workshop participants in Table 5 can be divided into three main themes: cascading impacts, disaggregated impacts, and impacts on marginalised communities, which we discuss in Sect. 4.2.2. Within this discussion section (Sect. 4.2.2), we explore the positive feedback loop between multi-hazard events, increasing informality, and the interrelationships between marginalisation and vulnerability in the Kathmandu Valley context in greater detail.

## 4 Discussion

In this discussion, we highlight five major themes. First, we consider the challenges in finding evidence of multi-hazard interrelationships in the blended source types (Sect. 4.1), detailing specific multi-hazard interrelationships and reasons for fewer multi-hazard interrelationship exemplars. Following this, we discuss findings from the workshop with practitioner stakeholders engaged in DRR strategies in the Kathmandu Valley (Sect. 4.2) and the single hazard and multi-hazard interrelationship impacts described in the blended evidence types (Sect. 4.3). We then consider the limitations within the methodology (Sect. 4.4). Finally, we outline the scalability of our methodology to other data-scarce urban settings (Sect. 4.5) and suggest future research directions (Sect. 4.6).

**Table 5.** List of 12 impact examples shared by participants on a virtual pinboard (Padlet) page during a practitioner stakeholder workshop. The table lists the number of impact examples, direct quotes shared by participants, and the hazard types causing the impacts.

| Impact example number | Direct quote shared by participant | Hazard types causing impacts |
| --- | --- | --- |
| 1 | "Communities along highways into valley – high exposure to EQ [earthquake] + landslide damage, but also dependent on traffic for livelihoods." – Kathmandu Workshop Participant A. | Earthquake and landslide |
| 2 | "Indirect impacts: still only limited capacity to respond to disasters at municipality or provincial level, so direction + materials must still come from capital. So if KTM [Kathmandu] is responding to an event, other parts of the country will have to wait." – Kathmandu Workshop Participant A. | All theoretically possible hazards |
| 3 | "Flood has a direct impact on urban poor mainly those living in a temporary shelter built in the bank of Bagmati river. Every year, flood[s] terrify those who are living in the informal settlements. These temporary households [are] also affected by windstorm. Assets damaged and or assets los[t] due to these hydro-climatic impacts have direct connection with livelihoods of the people." – Kathmandu Workshop Participant B. | Flood and storm (windstorm) |
| 4 | "Multi-hazard scenarios increase "uncertainty" which affect primarily migrants and marginalized dwellers." – Kathmandu Workshop Participant C. | All theoretically possible hazards |
| 5 | "Multi-hazards effect on land uses increasing inundation and landslides which affect farmers, women and labour." – Kathmandu Workshop Participant C. | All theoretically possible hazards |
| 6 | "Flooding of homes and businesses impacts society particularly those who are more vulnerable, [including] migrants." – Kathmandu Workshop Participant D. | Flood |
| 7 | "Landslides impact physical infrastructure, livelihoods, landscapes and increases uncertainty in people." – Kathmandu Workshop Participant D. | Landslide |
| 8 | "Migrants who cannot vote in KV's [Kathmandu Valley's] cities are particularly vulnerable to impacts." – Kathmandu Workshop Participant D. | All theoretically possible hazards |
| 9 | "Impacts of EQ [earthquake] or large monsoon on hydropower function and electricity supply to KTM [Kathmandu], via direct damage to infrastructure (shaking/landslides) or protracted sedimentation → impact upon wider power grid → impacts all users of power via load shedding." – Kathmandu Workshop Participant E. | Earthquake and storm (monsoon rain) |
| 10 | "Socio-economic impact, gender inequality, development challenge and challenge in meeting development goals. Increased vulnerability of ecosystem[s] and communities." – Kathmandu Workshop Participant F. | All theoretically possible hazards |
| 11 | "Direct Impact: human death, building damage, physical infrastructures such as road, bridge etc." – Kathmandu Workshop Participant G. | All theoretically possible hazards |
| 12 | "Jobs and livelihood, social-cultural and organizational impacts (e.g. system, ethics, Indigenous, organizations, traditional festiv[iti]es), health infrastructure, education, and micro-infrastructures." – Kathmandu Workshop Participant G. | All theoretically possible hazards |

## 4.1 Challenges in finding case study evidence of multi-hazard interrelationships that have influenced the Kathmandu Valley

We found 21 single hazard types and 83 potential multi-hazard interrelationships that have influenced or could potentially influence the Kathmandu Valley, of which 15 (71 %) of the single hazard types and 12 (14 %) of the multi-hazard interrelationships were evidenced by case studies. For multi-hazard interrelationships, we found only 21 sources of evidence, compared to the 58 sources of evidence for single hazards. These results support the findings of Gill et al. (2020) and Šakić Trogrlić et al. (2024b), which emphasise that there is a focus on detailed reporting and description of single hazards which is not found, both in research and understanding, for multi-hazard interrelationships.

This low proportion (14 %) of direct evidence for multi-hazard interrelationships is likely not due to lack of occur-

rence of multi-hazard interrelationships but rather a lack of documented multi-hazard events in the Kathmandu Valley and the fact that we only searched for evidence in English language sources. Within these exemplars of multi-hazard interrelationships, we identified differences in the number of sources available for the interrelationships of each primary hazard. For example, case studies of earthquakes triggering landslides, earthquakes triggering earthquakes, storms triggering landslides, and storms triggering floods were the most prevalent multi-hazard interrelationship pairs in the literature. Conversely, we found no case study evidence for earthquakes triggering volcanic eruptions or ground heave events triggering landslides. Similar to Šakić Trogrlić et al.'s (2024b) study in Nairobi, Kenya, and Istanbul, Türkiye, we found that many sources predominantly described single hazard events, with brief mentions of additional hazard types rather than explicitly documenting multi-hazard interrelationships. These challenges contribute to multi-hazard data scarcity in the Kathmandu Valley, specifically regarding cascading events and their impacts. The evidence base of single hazard and multi-hazard interrelationship exemplars in the Kathmandu Valley Single Hazards and Multi-Hazard Interrelationships Database (Thompson et al., 2024), gathered using the methodology, could contribute towards resolving this knowledge gap through evidence-based decision-making in the context of the Kathmandu Valley. Future work could build upon our database by developing our methodology from a series of Boolean searches to a systematic review, utilising data mining techniques (De Brito, 2021), searching non-online sources (e.g. archival material), and engaging in discussions with practitioner stakeholders, as Sect. 4.6 outlines.

The relative scarcity of multi-hazard interrelationships within our findings mirrors broader concerns the DRR community has expressed. Currently, there are three main databases that capture multi-hazard events across a regional to global scale: DesInventar Sendai, EM-DAT, and Munich Re. Recent studies have developed methodologies to systematically gather multi-hazard events into global multi-hazard datasets (e.g. Claassen et al., 2023; Jäger et al., 2024; Lee et al., 2024) to complement these existing disaster databases. On local to national scales, there are examples of multi-hazard interrelationship databases situated in Global South contexts, for example, Guatemala (Gill et al., 2020); the hydrological catchments of the Red River, Vietnam, and the Marikina Basin, the Philippines (Payo et al., 2022); Nairobi, Kenya, and Istanbul, Türkiye (Šakić Trogrlić et al., 2024b); and the Philippines (Ybañez et al., 2024). Our database contributes to this growing catalogue of multi-hazard event records and responds to calls for continued multi-hazard characterisation, the focus of many recent studies on multi-hazard interrelationships and impacts (e.g. De Angeli et al., 2022; Ward et al., 2022) to address knowledge gaps in multi-risk and its components (Šakić Trogrlić et al., 2022). Multi-hazard case studies may be available from grey literature

sources; however, these seldom go through the peer-review publication process (Šakić Trogrlić et al., 2024a) and may not be translated into additional languages.

Additionally, our database's broad coverage of hazard groups demonstrates how blended evidence sources can mitigate an overrepresentation of certain hazard groups, as documented by Owolabi and Sajjad (2023) in their review of multi-hazard risk analysis research published from 1994 to 2022. A lack of research across hazard groups has been exacerbated by a paucity of international collaboration between scholars from Global South regions, including South Asia and South America, despite higher exposure to multi-hazard impacts across many of these regions (Owolabi and Sajjad, 2023). This underrepresentation in research on multi-hazard events can be attributed to barriers to systematically collecting and documenting multi-hazard data, including information on interrelationships and impacts. These include governance-related factors, such as silos between organisations working on specific hazards and fragmentation in multi-hazard approaches (Šakić Trogrlić et al., 2024a). Limitations in resources and reduced institutional capacities also contribute towards a continued focus on single hazard and multi-layered single hazard approaches to data collection and documentation (Šakić Trogrlić et al., 2024a).

In the context of the Kathmandu Valley, a combination of numerous factors, including limited resources, public demands, fixed tenure, and bias towards physical infrastructure investment, contribute towards a deprioritisation in DRR activities (Poudel et al., 2021), including the continued shift towards multi-hazard DRR approaches. To address these challenges, we make the following tentative suggestions:

– continued communication beyond traditional disciplinary boundaries to combat siloed working and fragmentation

– enhancing regional collaboration to standardise multi-hazard event data collection and documentation strategies

– adaptation of existing frameworks to support the shift from single hazard and multi-layered single hazard to multi-hazard approaches

– augmenting existing support of digitised, open-access, multi-hazard event databases with the potential for language translation.

Examining our methodology's limitations (Sect. 4.4) and scalability (Sect. 4.5) is critical in mitigating these overarching issues in multi-hazards research.

## 4.2 Workshop findings

### 4.2.1 Multi-hazard interrelationship scenarios

The findings from our workshop, conducted with practitioner stakeholders engaged in DRR in the Kathmandu Valley, sup-

plemented the exemplars collated from blended sources of evidence (Sect. 3.1 and 3.2). The workshop provided a forum for participants to share their perspectives on multi-hazard interrelationship scenarios and multi-hazard impacts influencing Kathmandu Valley. The diversity in the subject backgrounds of participants provided a broad range of insights that complements the information collated from the blended evidence types (academic literature, grey literature, media, databases, and social media) described in Sect. 3.1 and 3.2 (Matanó et al., 2022). The discussions produced some novel findings concerning the hazardscape in the Kathmandu Valley, specifically in producing multi-hazard interrelationship scenarios for the context of the Kathmandu Valley by practitioner stakeholders working within DRR. These multi-hazard interrelationship scenarios complement existing literature that documents the variety of multi-hazard scenarios that have influenced the Kathmandu Valley and those that could theoretically influence the valley in the future (Gautam et al., 2021; Gill et al., 2021a; Khatakho et al., 2021).

The discussion of multi-hazard interrelationship scenarios centred around biophysical, geophysical, and hydrological hazard groups and simple triggering or increased-probability interrelationships. In both case studies and theoretical examples, participants described how anthropogenic processes increased the severity of multi-hazard impacts and altered response efforts following hazard events. A systematic risk assessment of multiple hazard types (earthquake, flood, landslide, and urban fire) in the Kathmandu Valley conducted by Khatakho et al. (2021) found that old settlements, densely populated settlements, and the central valley were the most risk-prone regions in the valley, supporting the participants' expert knowledge of the Kathmandu Valley context. Participant A commented on how the extension of development into the hills surrounding the Kathmandu Valley has increased the exposure of communities to multi-hazard impacts, particularly when coupled with heightened social vulnerabilities such as lower socio-economic status or marginalised identities. Indeed, urban poor communities and other marginalised groups experience heightened risk due to high exposure to multi-hazards and social vulnerabilities (Pelling et al., 2004). The multi-hazard interrelationship scenarios developed by participants provided further evidence for multi-hazard interrelationship pairs already given in the matrix for the Kathmandu Valley (Fig. 5). Additional scenarios that supplement those included in the matrix are Scenarios 9 (wildfire → air pollution), 10 ("haphazard development" + faulty electrics → urban fire), and 11 ("haphazard development" + soil conditions → ground displacement). Note that the participants were not restricted to discussing the hazards that are included in our multi-hazard interrelationship matrix (Fig. 5).

To expand upon the methodology developed by Gill and Malamud (2014), we included the co-production of multi-hazard interrelationship scenarios, similar to Šakić Trogrlić et al. (2024b), and additionally their impacts. Participants discussed how cascading impacts can interact dynamically across spatial and temporal scales and contribute to systemic impacts propagating across sectors. This systems thinking mirrors the development of Gill and Malamud's (2014) matrix by Matanó et al. (2022) to include socio-economic impacts. For example, the impact of earthquake or monsoon events on hydropower function and electricity supply has wide-reaching effects across the Kathmandu Valley due to load-shedding power cuts. This reduction in energy supply is likely to disproportionally impact urban poor communities due to less reliable electricity connections even before load-shedding activities. Within these communities, the burden of power outages is expected to be unevenly distributed owing to complex sociocultural factors (Bajracharya et al., 2022), emphasising the importance of a disaggregated and intersectional approach to multi-hazard impacts (Brown et al., 2019). Participants commented on the relationships between higher vulnerabilities of marginalised communities, notably the positive feedback loops between multi-hazard events and increasing informality. Of note is the increase in landslides following the 2015 Gorkha earthquake (e.g. as evidenced by Kargel et al., 2015, who reviewed satellite observations), contributing to increased informality within the Kathmandu Valley. These observations by participants reflect the informal conditions that many long-term internally displaced people (IDP) experienced whilst still living in camps and temporary accommodation in the urban periphery many years after the Gorkha earthquake (Titz, 2021).

Facilitating workshops to better understand multi-hazard cascades can be valuable for practitioner stakeholders considering emerging risks and future scenarios to inform decision-making processes (e.g. Riddell et al., 2019; Strong et al., 2020). Developing multi-hazard interrelationship scenarios from workshop participants and the multi-hazard interrelationship matrix (Fig. 5) can be helpful for hazard practitioners and agencies working in the DRR space. Applications include evaluating the effectiveness of preparedness and response systems, guiding land use planning, communicating educational messages towards at-risk communities, and facilitating dialogue between practitioner stakeholders and at-risk communities (Gill et al., 2020). In the Nepali context, developing and quantifying multi-hazard scenarios would support preparedness and recovery strategies and the allocation of resources on provincial and national scales (Gautam et al., 2021).

### 4.2.2 Impact examples

The impact examples shared by participants on the Padlet page (Table 5) can be subdivided into the following themes:

– cascading impacts,

– disaggregated impacts, and

– impacts on marginalised communities.

Cascading impacts, or networks of interdependent impacts, may be dynamic and change over space and time (De Brito, 2021). They also occur as part of broader systems that emphasise feedback between impacts (Spoon et al., 2020; Hochrainer-Stigler et al., 2023). One participant commented on how the "impacts of earthquake or large monsoon on hydropower function and electricity supply to KTM [Kathmandu], via direct damage to infrastructure (shaking/landslides) or protracted sedimentation, impact upon wider power grid [which] impacts all users of power via load shedding". As noted in Sect. 4.2.1 this observation demonstrates how direct impacts of a hazard event can have broader systemic effects that influence communities across greater spatial and temporal scales than the hazard event itself.

Another theme in the Padlet pages is the disaggregation of multi-hazard impacts. In this case, we define disaggregated impacts by social group (e.g. gender, age, socio-economic status, disability). One participant noted that we must think beyond direct tangible impacts to consider "multi-hazards effect on land uses, increasing inundation and landslides which affect farmers, women and labour". Effects such as "assets damaged and or assets los[t] due to these hydro-climatic impacts has direct connection with livelihoods of the people" contribute towards anxiety and a chronic state of emergency experienced by urban poor communities. The consideration of indirect and intangible impacts is necessary when addressing "socio-economic impact, gender inequality, [and] development challenge[s]".

Considering marginalised communities was a closely related theme to disaggregated impacts. Participants emphasised the increased vulnerability of urban poor communities who live in temporary accommodation on riverbanks: "flood has a direct impact on urban poor, mainly those living in temporary shelter built on the bank of Bagmati river. Every year, flood terrif[y] those living in the informal settlements." Participants also noted the vulnerability of other marginalised groups, where "multi-hazard scenarios increase 'uncertainty' which affect primarily migrants and marginalised dwellers" and "migrants who cannot vote in the Kathmandu Valley's cities are particularly vulnerable to impacts".

Despite increasing focus on cascading and disaggregated impacts, there remain gaps in multi-hazard interrelationship knowledge, including a detailed understanding of the direct and indirect impacts of multi-hazards necessary for effective mitigation (Šakić Trogrlić et al., 2024a). A potential extension of the workshop with practitioner stakeholders is to incorporate questions that consider variables of vulnerability and impact into the multi-hazard interrelationship matrix. Existing multi-hazard visualisations (e.g. Gustafsson et al., 2023) illustrate the interrelationships and impacts resulting from hazard cascades and provide potential approaches to incorporating these variables within our methodology. For example, Sharma et al. (2023) illustrate hazard cascades and their impacts in the central Himalayas, including the duration, scale, and sector influenced by each hazard cascade

event using a multi-hazard interrelationship matrix similar to Fig. 5 in our paper.

Another important consideration in visualising the multi-hazard interrelationship matrix is the useability of the tool by practitioner stakeholders engaged in the hazardscape region. When incorporating the tool into existing DRR strategies, understanding the spatial and temporal components of multi-hazard events is critical in coordinating an appropriate and tailored response. For example, De Angeli et al. (2022) developed a multi-hazard risk framework for spatial–temporal impact analysis and applied it to a seismic and flood damage scenario in the Po Valley, Italy. The spatial and temporal evolution of the multi-hazard event scenario is visualised with the following components: hazard maps at various time instants, the temporal evolution of hazard impacts, and the impacted area.

Both Sharma et al. (2023) and De Angeli et al.'s (2022) visualisations present potential approaches to expanding our existing multi-hazard interrelationship matrix in the Kathmandu Valley context. Including impact variables within Fig. 5 would enhance our methodology's scalability and utility within DRR strategies (discussed further in Sect. 4.5 and 4.6). A clear visualisation of the evolution of impacts across type and spatial–temporal extent in a figure or series of figures would be helpful as a dissemination tool in decision-making processes.

## 4.3 Single hazard and multi-hazard interrelationship impacts

While we compiled literature on evidence for single hazard types and multi-hazard interrelationships influencing the Kathmandu Valley, we also noted the impacts described. Within the exemplars we collated from blended source types, as documented in the Kathmandu Valley Single Hazards and Multi-Hazard Interrelationships Database (Thompson et al., 2024), we found the most detailed descriptions of single hazard (Column 9, Sheet "A. Single Hazards Evidence") and multi-hazard interrelationship (Column 6.1, Sheet "B. Hazard Interrelationships Evidence") impacts in academic literature and grey literature (e.g. UNDRR reports). Media and social media (e.g. YouTube videos) also provided informative accounts, but descriptions often described generic and larger spatial impacts rather than event-specific ones. The depth of information on impacts usually reflected the typical frequency of the hazard and/or its level of impact. For example, information about the impacts of extreme temperature (cold) events was limited to generic descriptions of environmental and socio-economic consequences due to the rare occurrence of low temperatures in the valley. Most impact information was centred on direct quantitative information, such as infrastructure damage, injuries, and loss of life. For instance, descriptions of ground collapse impacts were limited to general information about fatalities and disruption. The limited number of indirect and intangible impacts could be due to

sampling bias of the source types used in the study, the sample size of the hazard events impacting the Kathmandu Valley, and information accuracy when verification of source types is not possible (Matanó et al., 2022).

When indirect and qualitative impacts of single hazard events were documented, the common themes were the disproportionate burden some social groups experienced (disaggregated impacts) due to variable exposure, vulnerability, and anthropogenic processes. For example, following the 2015 Gorkha earthquake, aftershocks (and resulting landslides) contributed to approximately 2.5 %–3.5 % of the national population entering poverty – equivalent to 700 000 additional economically disadvantaged people (ILO, 2017) – where low-caste and poorer communities experienced the greatest severity of impacts due to "marginalised status, limited resources and livelihood options" (UNDRR, 2019). These impacts emphasised the relationship between communities' socio-economic status and vulnerability. The disproportionate burden on some social groups was echoed in the reporting of drought events impacting the Kathmandu Valley (IIED, 2010). Long-term drought in the early 2000s resulted in gendered consequences where missed education significantly affected girls and increased water theft from neighbouring wells and water trenches (IIED, 2010). The impacts of drought, exacerbated by overpopulation and rapid urbanisation, may undermine social cohesion as water shortages increase the likelihood of conflict between communities in the valley in the future (Adhikari, 2019).

Less impact information was documented for multi-hazard events, perhaps since fewer details of multi-hazard interrelationships were recorded across all source types. As a result of the 2015 Gorkha earthquake and aftershocks, over 50 % of fatalities were of individuals from marginalised communities. For example, Tamang communities experienced disproportionate impacts of the event due to "poverty, neglect and outright discrimination" (Magar, 2015). Higher-magnitude multi-hazard events generally included a greater breadth and depth of impact information as these events have more significant spatial and temporal impacts and are more likely to be documented across source types. For example, storm-triggered flood events in 2019 increased the occurrence of diseases like dengue fever (Molden and McMahon, 2019). Further storm-triggered floods and landslides in 2021 disproportionately affected urban poor communities, owing to the most significant damage occurring in low-lying informal settlements (ReliefWeb, 2021).

Incorporating single hazard and multi-hazard interrelationship impacts into our methodology extends our work from previous studies that focused primarily on the hazard component. By incorporating broader aspects of the Kathmandu Valley hazardscape, we emphasise the importance of an interdisciplinary approach to DRR research. In doing so, our work responds to calls for hazard scientists to continue integrating a diverse range of data sources to support a more nuanced understanding of multi-hazard scenarios and broader hazardscapes (Gill et al., 2021b).

## 4.4 Limitations

We recognise that limitations in our methodology may have altered the types and relative quantity of specific hazard events, impacts, and multi-hazard interrelationship scenarios we observed in our results. This section highlights limitations in (a) the collation of blended sources of evidence and (b) the practitioner stakeholder workshop.

### 4.4.1 Limitations in the collation of blended evidence sources

Two main factors contributed to uncertainty during the systematic approach to selecting evidence:

– We used a *limited number of keywords* during the search process, thus limiting the number of publications returned; alternative keywords would have yielded different results. This limitation includes variations on hazard terms such that different spatial or temporal terminology versions do not limit the number of returned publications (Taylor et al., 2015).

– We searched for evidence using *English language* databases, search engines, and media websites. Solely conducting searches in English reduced the number of publications returned whilst also losing the nuance and context of single hazard types and multi-hazard interrelationships described in Nepali language publications (Šakić Trogrlić et al., 2024b).

We minimised these limitations by considering efficient and practical solutions for each source of uncertainty. It would be impractical to include a long list of keywords during the search process; instead, we included three to four specific words to target the most relevant publications for each single hazard or multi-hazard interrelationship. For instance, "Kathmandu AND storm* AND flood* AND impact*" identified examples of storm-to-flood hazard sequences without specifying the type of multi-hazard interrelationship, which may have excluded the return of some publications (Taylor et al., 2015). Searching in three reputable online English language Nepali newspapers reduced the English language limitation. By searching across online newspapers, a greater breadth and depth of sources could be returned than by solely using one newspaper whilst also returning publications detailing events across greater spatial and temporal scales (De Brito et al., 2021). We focused on publications from 2010 onwards to outline recent hazard events whilst not excluding low-probability high-impact events. This decision enables exemplars to be viewed in the current context regarding multi-hazard knowledge and approaches to DRR.

### 4.4.2 Limitations in the workshop with practitioner stakeholders

The findings from the workshop represent a snapshot of the hazardscape in the Kathmandu Valley. They are a product of the perspectives and identities of those present in the discussion and those absent (Leonard et al., 2014), with findings potentially affected by professional interests, lack of gender balance, and researcher positionalities. Regarding participants' professional interests, three specialised in the knowledge of multi-hazards from a physical sciences perspective, three in interdisciplinary approaches, and one in understanding risk from a social sciences context. Although we approached participants with a range of subject expertise (Matanó et al., 2022), due to availability, there was a bias towards landslide and earthquake hazards, cascades, and impacts, as three of the participants had expertise in these fields.

As described in Sect. 2.4 the ratio of Nepali or Nepali-based to British or British-based participants was designed to minimise the effect of power asymmetries within the discussion and create an atmosphere where all participants felt able to share their perspectives (Secor, 2010; Wolf, 2018). Conversely, the gender balance between participants was less representative despite approaching approximately equal numbers of female and male participants. To minimise this imbalance, we aimed to facilitate the session in a manner that decentred our role as facilitators and limited control of the conversation by one or a few individuals. Our positionalities as researchers may have affected the discussion dynamics, particularly regarding what information was shared or withheld, how participants described case studies and theoretical examples, and what details they included. By inviting participants with whom we have working connections and partnerships or are within our research network, we hoped to share knowledge built on these sustainable connections and a greater sense of trust (Wilmsen, 2008). The following section examines the scalability of our methodology to other data-scarce urban contexts.

### 4.5 Scalability to other data-scarce urban settings

Building upon previous work (Gill et al., 2020; Matanó et al., 2022; Gustafsson et al., 2023; Šakić Trogrlić et al., 2024b), this study has furthered existing methodologies to collate blended sources of evidence of single hazard types, multi-hazard interrelationships, and their impacts. We demonstrate that it is possible to systematically gather case studies and theoretical examples of multi-hazard events to improve knowledge of hazardscapes in data-scarce urban settings. This challenge is particularly relevant in urban settings in low- to middle-income countries (Osuteye et al., 2017).

With application to the Kathmandu Valley, we have developed this methodology to collate single hazard and multi-hazard event data on finer spatial resolutions (e.g. ward level within the Kathmandu Valley) than in previous studies, as well as impact data. We have achieved this finer spatial resolution by using a systematic review of blended sources of evidence (academic literature, grey literature, media, databases, and social media) to minimise the effect of data scarcity and provide evidence from various perspectives. The workshop component of the methodology enables practitioner stakeholders engaged in DRR work in the Kathmandu Valley to co-produce multi-hazard interrelationship scenarios and their impacts. This knowledge generation supplements the blended evidence sources we collated for single hazard types and multi-hazard interrelationships and emphasises the most significant scenarios and impacts in the Kathmandu Valley context. These multi-hazard interrelationship scenarios can support dialogue between practitioner stakeholders engaged in people-centred DRR strategies in the local context, raise awareness in at-risk communities, support risk-sensitive land use planning, and strengthen hazard preparedness and response strategies (Gill et al., 2020, 2021a).

Applying learnings from the Kathmandu Valley context, this methodology has scalability to other data-scarce urban settings as it utilises a variety of blended source types, given their availability. Once the researcher applies spatial and temporal boundaries to the chosen study area, they can use systematic searches (Sect. 2.1) to gather theoretical and case study events across blended source types, and workshop considerations outlined in Sect. 2.4 to co-produce results with groups responsible for managing risk in that location. Although it may be appropriate to focus on more recent hazard events to capture the current state of the hazardscape, searching across a broader temporal range would enable an analysis of how patterns in multi-hazard events and their impacts change across space and types of interrelationships. The researcher should consider what resolution is possible for the urban context chosen and how data scarcity affects which single hazard types, multi-hazard interrelationships, and impacts are returned in searches (Matanó et al., 2022). The methodology should be regularly assessed to gauge necessary improvements and apply recommendations (e.g. using a theory of change) (Gill et al., 2021b).

Our methodology's applicability to other geographical contexts, communities, and scales represents one aspect of its value as a "useful, usable and used" tool (Boaz and Hayden, 2002). On local and regional scales, the multi-hazard interrelationship matrix (Fig. 5) supports discussions between practitioner stakeholders, including hazard managers, policymakers, academics, NGO practitioners, and members of at-risk communities, on multi-hazard preparedness and planning. The methodology can act as a bridging tool between communities (Gill et al., 2020) to support the continued and required shift from single hazard and multi-layered single hazard approaches towards multi-hazard strategies (Ward et al., 2022). Breaking down silos between organisations engaged in DRR work is critical in working towards effective preparedness planning and mapping future multi-hazard scenarios (Scolobig et al., 2017; Gill et al., 2021b). This action

could involve the inclusion of multi-hazard interrelationships and impacts within existing DRR training materials and educational frameworks for locally situated learning or the development of effective multi-hazard early warning systems (MHEWS).

On an international level, developing this study's Kathmandu Valley Single Hazards and Multi-Hazard Interrelationships Database (Thompson et al., 2024) could help inform and enrich existing international disaster datasets (e.g. DesInventar Sendai, EM-DAT). Populating current databases with further examples of multi-hazard events and impacts can broaden evidence bases, particularly in more data-scarce regions, for use in funding applications, awareness raising of multi-hazard interrelationships, and developing preparedness plans. In the UK, a review of the National Security Risk Assessment (NSRA) methodology identified multi-hazard interrelationship frameworks as being a valuable tool to explore interdependencies, aiming to improve risk assessment practice to support planning processes and inform policy (Royal Academy of Engineering, 2023). Systematic multi-hazard information can complement new multi-hazard datasets, particularly those focused on urban contexts such as the MYRIAD-EU project's VulneraCity database of urban vulnerability drivers (Stolte et al., 2024), and centre the need for dynamic multi-hazard approaches to disaster risk.

## 4.6 Future work

In the future, we suggest that researchers engage further with local practitioner stakeholders and at-risk communities to collect critical insights into their main concerns regarding multi-hazards and DRR strategies. This engagement would support the implementation of the methodology we have outlined in this paper in the Kathmandu Valley context as a pilot for other data-scarce urban areas.

To this end, we included an additional column for hazard impacts in the Kathmandu Valley Single Hazards and Multi-Hazard Interrelationships Database (Thompson et al., 2024) to allow for further context and insight into the consequences of each exemplar. We included a column for input from practitioner stakeholders (Column 7, Input from practitioner stakeholders, Sheet "B. Hazard Interrelationships Evidence") to assess the following:

- Is the identified interrelationship relevant for/applicable to the Kathmandu Valley?

- Would you classify the identified interrelationship as important for today's Kathmandu Valley?

- Is this interrelationship relevant for the future Kathmandu Valley (e.g. will become increasingly important), and should it be considered in urban planning?

We listed a final column for practitioner stakeholder prioritisation (Column 8.1, Input from practitioner stakeholders – prioritisation, Sheet "B. Hazard Interrelationships Evidence"), asking practitioner stakeholders to

- indicate the most critical multi-hazard interrelationships in today's Kathmandu Valley

- list multi-hazard interrelationships that you feel will be relevant for tomorrow's Kathmandu Valley (including the interrelationships which are already relevant today).

Providing an up-to-date summary of single hazard types influencing the Kathmandu Valley, or with a theoretical chance of occurrence, could give practitioner stakeholders more detailed insight into the natural hazards influencing the valley. These practitioner stakeholders include government agencies (e.g. Ministry of Home Affairs, MOHA), non-governmental organisations (e.g. Practical Action Nepal; Lumanti Support Group for Shelter), academia (e.g. Tribhuvan University), and the private sector (e.g. Atullya Foundation Pvt. Ltd.) working on inclusive approaches to DRR in the Kathmandu Valley and Nepal more widely.

Additionally, including anthropogenic processes within the methodology can add nuance to the breadth of single hazard types and multi-hazard interrelationships across geographical contexts. Here, we understand anthropogenic processes as "intentional, non-malicious human activities" as defined by Gill and Malamud (2017). In their research, Gill and Malamud (2017) present a systematic literature review of past research on anthropogenic processes, focusing on categorising artificial ground and land use. Following this, they characterise the role of anthropogenic processes in triggering natural hazard events and catalysing or impeding the interrelationships between natural hazards (Gill and Malamud, 2017). Moving forward, we recognise the need to develop our methodology further to include anthropogenic processes as an integral component of any hazardscape.

Another potential expansion is the inclusion of non-natural hazards, including the following categories as defined in the UNDRR-ISC hazard information profiles (Murray et al., 2021):

- *Biological hazards* are a broad range of hazards of "organic origin".

- *Chemical hazards* include human exposure to chemicals of human and natural origin.

- *Environmental hazards* include degradation of "natural systems and ecosystem services".

- *Societal hazards* are "human activities and choices" that present risks to communities and environments.

- *Technological hazards* include failure of existing and emerging technology, impacting both within and outside systems.

Although our methodology has focused on natural hazard types, it can be applied to a broader dataset, including additional hazard types and spatial and temporal scales, to capture hazardscapes across different geographical contexts. Natural hazards are situated within the wider systems in which they occur and can be directly or indirectly influenced by anthropogenic processes (Gill and Malamud, 2017), particularly in urban contexts. It is our aspiration that integrating anthropogenic processes and additional hazard types into the existing methodology and multi-hazard interrelationship matrix results in a more nuanced understanding of urban hazardscapes. These elements will extend existing multi-hazard scenarios to incorporate risk variables. This knowledge is significant for practitioner stakeholders working in specific urban contexts to evaluate which components of dynamic risk scenarios can be targeted to reduce impacts on at-risk and marginalised communities.

## 5   Conclusions

This paper has detailed the systematic and evidenced process of collating exemplars of single hazard types and multi-hazard interrelationships in a data-scarce urban area with application to the Kathmandu Valley. We supplemented these exemplars with the perspectives of practitioner stakeholders engaged in DRR in the Kathmandu Valley context. Using blended evidence types increases the depth of information about multi-hazard interrelationship scenarios and their impacts (Neri et al., 2008; Gill et al., 2020; Taylor et al., 2020). Data-scarce urban areas in the Global South represent an important research focus for deepening understanding of multi-hazard interrelationship scenarios and impacts. The high vulnerabilities of urban poor communities, combined with rapidly increasing population growth and exposure to complex multi-hazard interrelationships (Dodman et al., 2022), present a significant challenge to effective community-led DRR strategies. Improved knowledge of multi-hazard interrelationships and cascades in the Kathmandu Valley would enable a more holistic approach to DRR from various practitioner stakeholders. We argue that this paper promotes the use of blended evidence types in collating single hazard and multi-hazard event information in data-scarce urban settings. It demonstrates the importance of disaggregated impact information in supporting communities to respond to hazards and increase their resilience to future events. We suggest that the methodology presented in this paper could contribute towards resolving some of these data obstacles across data-scarce urban regions (UNDRR, 2023).

Across our systematic searches, exemplars of single hazards were more prevalent across multiple source types (i.e. academic literature, grey literature, media, databases, and social media). There was greater detail of the anthropogenic processes driving the hazards and impacts for single hazard exemplars than the multi-hazard interrelationship exemplars.

The 58 single hazard sources we selected from the literature search evidenced 21 single hazard types that might influence the Kathmandu Valley. Discussions with workshop participants supported the single hazard types documented in the blended source types. In contrast, searching for evidence of triggering and increased-probability multi-hazard interrelationships in the Kathmandu Valley was challenging across all source types. We selected 21 multi-hazard interrelationship sources, which evidenced 12 specific multi-hazard interrelationships that might influence the Kathmandu Valley out of 83 we propose to have a theoretical possibility of influencing the valley. Workshop participants confirmed the challenge of documenting complex multi-hazard interrelationships by predominantly describing primary to secondary multi-hazard interrelationships. The discussion also developed the process of considering multi-hazard impacts in the Kathmandu Valley, specifically knowledge of cascading impacts, disaggregated impacts by social group, and the disproportionate impact borne by marginalised communities (Brown et al., 2019; Dodman et al., 2022).

Our Kathmandu Valley Single Hazards and Multi-Hazard Interrelationships Database (Thompson et al., 2024) can support assessing which hazard types and multi-hazard interrelationships are most significant in the Kathmandu Valley and how this may change. Local practitioner stakeholders can integrate a better understanding of the hazardscape and resulting impacts in the Kathmandu Valley into more holistic approaches towards multi-hazard DRR. We suggest that this paper's findings could contribute to developing multi-hazard interrelationship scenarios as part of the objectives of the midterm review of the Sendai Framework (UNDRR, 2023) and other national and international research priorities.

*Data availability.* The blended sources used to collate the single hazard types and multi-hazard interrelationships in the Kathmandu Valley are available in the Kathmandu Valley Single Hazards and Multi-Hazard Interrelationships Database. This database was compiled using the methodology described in this paper and is available in the online open-access repository Zenodo (https://doi.org/10.5281/zenodo.13749299) with CC-BY 4.0 international rights, cited as Thompson et al. (2024).

*Author contributions.* HET, JCG, RST, and BDM conceptualised the research and developed the methodology. HET analysed the data, undertook the investigation, visualised the results, and wrote the manuscript draft. JCG, RST, FET, and BDM supervised the research. All co-authors reviewed and edited the manuscript.

*Competing interests.* At least one of the (co-)authors is a member of the editorial board of *Natural Hazards and Earth System Sciences*. The peer-review process was guided by an independent editor, and the authors also have no other competing interests to declare.

ther geographical representation in this paper. While Copernicus Publications makes every effort to include appropriate place names, the final responsibility lies with the authors.

*Special issue statement.* This article is part of the special issue "Methodological innovations for the analysis and management of compound risk and multi-risk, including climate-related and geophysical hazards (NHESS/ESD/ESSD/GC/HESS inter-journal SI)". It is not associated with a conference.

*Acknowledgements.* We are grateful to the seven practitioner stakeholder workshop participants described in this paper for their rich and detailed insights into the hazardscape of the Kathmandu Valley. This work was supported by the Department of Geography, King's College London; the Tomorrow's Cities research hub; and the British Geological Survey. CE4

*Financial support.* This research has been supported by the Department of Geography, King's College London's in-kind contribution to the Tomorrow's Cities Research Grant (UKRI/GCRF fund under grant no. NE/S009000/1) and the British Geological Survey University Funding Initiative (BUFI) PhD Studentship (S466).

*Review statement.* This paper was edited by Marleen de Ruiter and reviewed by Nathan Clark and one anonymous referee.

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

**Remarks from the language copy-editor**

CE1    Please note that slight adjustments are fine during proofreading, but the removal or addition of information would require the approval of the handling editor.

CE2    The requested adjustment needs to be approved by the handling editor.

CE3    Please note that it is our house standard only to capitalize proper nouns.

CE4    Could this last sentence not be removed? The same information is given below.

**Remarks from the typesetter**

TS1    Please confirm the change.

TS2    Please prepare an explanatory document (pdf) which we can send to the editor via our system.

TS3    Please add this reference to the main text.