# Peer review of "Revision 1: nhess-2024-101"

_Natural Hazards and Earth System Sciences, 2024_

## Author Comment (AC1)

**Reply to reviewer comments on NHESS-2024-101**

We express our sincere thanks to both reviewers (R1 and R2) for their time, insights and constructive engagement with our manuscript **"A methodology to compile multi-hazard interrelationships in a data-scarce setting: an application to Kathmandu Valley, Nepal"** by Harriet E. Thompson, Joel C. Gill, Robert Šakić Trogrlić, Faith E. Taylor, and Bruce D. Malamud, submitted on 28 May 2024 to *Natural Hazards and Earth System Sciences* (Manuscript NHESS-2024-101).

Below, we provide detailed responses to the reviewers' comments of how we will proceed if the editor invites us to submit a revision. Line numbers refer to those in the original version of the manuscript, and we have added additional references to the reference list. We believe that the suggested changes from both reviewers have improved the quality and argument of the manuscript and thank the reviewers again for their contributions.

**Reviewer 1 (R1) (Nathan Clark)**

**(R1-00):** "The paper provides a very interesting and timely methodological approach for identifying and compiling information on interrelationships among single and multi-natural hazards. Overall the paper is well written and well positioned for a place within the Special issue: Methodological innovations for the analysis and management of compound risk and multi-risk, including climate-related and geophysical hazards. Some minor considerations for the authors:"

**(Reply to R1-00)**: We thank R1 for their comments on the interesting and timely approach for compiling information. Below we respond to each comment in turn.

**Introduction:**

**(R1-01):** "Overall this section is clear and concise, however, a few parts of the narrative could flow better and need clarification.

In terms of data-scarcity on hazard impact data, and this quote: "One of the significant challenges facing urban areas in Nepal is the scarcity of hazard impact data, which is a barrier to effective DRR strategies (Chatterjee et al., 2015; SIAS, 2016)." The first thought that came to my mind is wondering the extent which this has changed almost 10 years since the 2015 Earthquake and from the sources referenced. And if we are talking about scarcity of multi-hazard impact data, I wonder how Kathmandu differs so much from other urban settings, even in the West.""

**(Reply to R1-01):** The quote above will be amended to the following:

"One of the significant challenges facing urban areas in Nepal is the scarcity of hazard impact data (Chatterjee et al., 2015; SIAS, 2016), except for a few notable examples."

The following text will be added to the revised manuscript to Section 1 Introduction:

"In the Nepali context, previous research and building of databases, by academics, government organisations and non-governmental organisations (NGOs) has centred on the impacts of single hazards (Bhatta and Adhikari, 2024). In response to recent multi-hazard events such as the Melamchi debris flow in 2021, part of a hazard cascade that caused multi-sectoral impacts on a regional scale across one year (Sharma et al., 2023), more studies are

focused on multi-hazard impact and risk assessments (e.g., Dunant et al., 2024). Nepali hazard event databases, including the Nepal Disaster Risk Reduction (DRR) Portal (2024) and Building Information Platform Against Disaster (BIPAD) Portal (Youth Innovation Lab, 2020), predominantly document direct and tangible impacts, with some basic disaggregation by gender. Similar to geographical contexts globally, across national income level and "Global North and South" categorisation, one of the significant challenges facing urban areas in Nepal is the scarcity of disaggregated multi-hazard impact data, which is a barrier to effective DRR strategies (Youth Innovation Lab, 2020; De Maio et al., 2024)."

**(R1-02):** "On lines 84 and 95 the authors note the Kathmandu experiences many single and multi-hazards events without qualifying it. Maybe the examples under 110-120 could be brought up under points 84 and 95 to create a more coherent narrative."

**(Reply to R1-02):** Examples of natural hazard events and cascades of further hazards or impacts in Nepal will be moved from L110-L120 to L89. After noting that Kathmandu Valley experiences a variety of single natural hazard and multi-hazard events, we will include the proposed paragraph in our Reply to R1-03.

**(R1-03):** "From lines 95 down, it is not immediate clear why the authors are highlighting the squatter settlements, and even including a visual. Is it only to illustrate population growth and growing urbanisation, or is it related to the heightened risks and impacts on vulnerable groups? This is connected indirectly to the Urban Fire bullet point, and then again later made clearer in the Results and Discussions sections with the focus on impacts to marginalised communities. I suggest the authors make these connections more clear in the Introduction when introducing the section on squatter settlements."

**(Reply to R1-03):** We will expand the paragraph which introduces squatter settlements in Kathmandu Valley. This paragraph will include an explanation for the increased vulnerability and disproportionate burden of impacts experienced by members of "marginalised" groups. Please see as follows our initial proposed text:

"Within this population, "marginalised" communities, including residents of squatter settlements within the broader grouping of the urban poor, experience a disproportionate burden of hazard impacts due to their heightened socio-economic vulnerability (Pelling et al., 2004; Gorman-Murray et al., 2018; Dodman et al., 2022). In mainstream hazard impact data, there is often a lack of representation of more vulnerable groups. Addressing this data gap is crucial to prevent Disaster Risk Reduction (DRR) strategies from unintentionally exacerbating existing social inequalities (Brown et al., 2019). To illustrate the spatial distribution of socially marginalised communities in one region of Kathmandu Valley, Fig. 1 presents six squatter settlements, with a total population of 7270 people in 2019 (Khanal and Khanal, 2022), out of 53 squatter settlements that are documented in the valley (DUDBC, 2010)."

**Methodology:**

**(R1-04):** "Only using English sources is certainly one of the main weaknesses of the study. But it is very good that the authors have recognised and highlighted the limitation in a number of places within the document.

What remains unclear is what is meant by "grey literature". Perhaps I missed it, but I find only one small explanation on line 608 which references "e.g., UNDRR reports". Did the authors include in their literature search any other NGO/IGO and government agency reports, and perhaps preparedness plans at national/regional/local/community levels? They could likely find more hazard specific frequency magnitude/impact information through specific agencies in the regions, for instance NSET for geophysical hazards. https://www.nset.org.np/nset2012/. These types of sources seem very relevant to have been included considering the scope of the paper,"

**(Reply to R1-04):** We will include Auger's (1998) widely accepted definition of grey literature. In addition, we will list further subtypes (on line 196) of grey literature that we selected for inclusion in the single hazard and multi-hazard interrelationships database.

**(R1-05):** "Line 214 notes: "We used the methodology described in the literature review in Sect. 2.1 to conduct Boolean searches for single hazards and multi-hazard interrelationships that might occur in Kathmandu Valley." Did this also include events which have already occurred? I suggest to make that clear."

**(Reply to R1-05):** We will amend this sentence to read: "We used the methodology described in the literature review in Section 2.1 to conduct Boolean searches for single hazards and multi-hazard interrelationships that have influenced or could potentially influence Kathmandu Valley."

**(R1-06):** "I appreciate the thought behind pulling out the column headings into separate boxes for ease of reading in Figures 2 and 3, but I think it might create even more difficulty for the reader. Or at least it makes the presentation of the Figures look quite messy. Consider if those extra boxes are really needed. Perhaps instead just highlight the relevant text in different colours in the tables."

**(Reply to R1-06):** Thank you for this comment. We will experiment with this suggested visualisation to improve the clarity of Figures 2 and 3 for the reader.

**(R1-07):** "Line 293 notes: "These participants were selected based on their in-depth knowledge of the Kathmandu Valley context and existing connections built on pre-established working relationships." What does this say about their specific backgrounds and expertise contributing to this topic? Especially given the small number or participants, this clarification is needed in the methodology section. I found the explanation (in part) only at the end of the document in the Limitations section."

**(Reply to R1-07):** To provide clarification, we will add the following sentences to this section (Section 2.4):

"Participants worked for organisations spanning the non-governmental organisation (NGO) or international non-governmental organisation (INGO) (e.g., Practical Action Nepal), national society (e.g., National Society for Earthquake Technology – Nepal, NSET), research institute (not listed due to consent form response), and academic (e.g., Durham University) sectors. We approached participants based on their expertise in single and multi-hazard interrelationships and their impacts in Kathmandu Valley, as applied from their perspectives as physical, social and interdisciplinary practitioners and/or researchers."

We will also add additional information on participants' research and/or practitioner backgrounds, where participants' responses to the pre-workshop consent form permit.

**Results:**

**(R1-08):** "Figure 5 is great! It also made me quite curious about some of the relationships. For instance I had to look up how a hailstorm can trigger a volcanic eruption.

Line 407 notes: "Additionally, we shared Fig. 6 with workshop participants to illustrate the value of the multi-hazard interrelationship matrix in extracting relevant multi-hazard scenarios." It is not clear if they were asked for feedback on that Figure, or if it was also used as a tool in the workshop? E.g. What did they think of it? Was it useful?"

**(Reply to R1-08):** Although it was not within the workshop's scope to ask for participant feedback on the multi-hazard interrelationships matrix itself (other than introducing them to it), we did use **Fig. 6** as a tool for participants to explore potential multi-hazard scenarios that could influence Kathmandu Valley. We will add the sentences as follows:

"Additionally, we shared **Fig. 6** with workshop participants to illustrate the value of the multi-hazard interrelationship matrix in extracting relevant multi-hazard scenarios. **Figure 6** was then made available virtually to participants during workshop discussions and acted as a visualisation tool and talking point to explore further multi-hazard interrelationships that have influenced or could influence Kathmandu Valley, beyond those archetypal examples that participants may associate with the region."

**(R1-09):** "Line 462 notes: "Including anthropogenic hazards and related processes could form the basis of future developments of this work." That seems like a fairly significant outcome, and should probably be highlighted in sections 4.6 and 4.7 on future work in this direction."

**(Reply to R1-09):** We agree with the reviewer that this should be highlighted as it is a part of our future research we are excited about pursuing. We are in discussion regarding how to expand upon this sentence in Sections 4.6 ("Scalability to other data-scarce urban settings") and 4.7 ("Future work"). This will include the following:

- We will present a definition and discuss a systematic review of anthropogenic processes and their influence on natural hazards done by Gill and Malamud (2017), both of whom are part of this current manuscript, with the desire to extend the findings from that work to dynamic risk scenarios.
- We will consider changing anthropogenic hazard to read anthropogenic processes (as in Gill and Malamud, 2017).
- We will outline additional hazard types that might occur. For example, biological, chemical, environmental, societal, technological — as described in the recent United Nations Hazard Information Profiles (Murray et al., 2021).
- We will discuss how anthropogenic processes can extend multi-hazard dynamic risk scenarios.

**Discussion:**

**(R1-10):** "In section 4.3.2. It could be interesting for the authors to reflect on HOW the different types of variables related to vulnerabilities and impacts (also within different spatial and

temporal scales) could be considered within such a matrix. The authors do provide some considerations for this in sections 4.6 and 4.7, maybe there is a way to also provide a few of these reflections already in 4.3.2."

**(Reply to R1-10):** In Section 4.3.2 ("Impact examples"), we will examine potential approaches of incorporating variables of vulnerability and impacts into the existing multi-hazard interrelationship matrix. Discussion points will include:

- Mention of some of the existing visualisations of multi-hazard events, their impacts and variables of vulnerability.
    - For example, Figure 1 in Sharma et al. (2023) shows hazard cascades and their impacts in the central Himalayas, including the duration, scale and sector affected by each hazard cascade event.
- Outline of a couple of potential approaches to incorporate spatial and temporal components into our existing multi-hazard interrelationships matrix.
    - For example, discussing the work of De Angeli et al. (2021) which has a framework for spatial-temporal impact analysis for multi-hazards and has a figure describing various scenarios for spatial-temporal overlaps of impact.
- Reflect on the considerations already mentioned in Sections 4.6 ("Scalability to other data-scarce urban settings") and 4.7 ("Future work") in Section 4.3.2 ("Impact examples").

**(R1-11):** "4.6 and 4.7 are really nice to read. Two things that do however seem to be missing (or not explicitly enough) in terms of scalability of the matrix: should scalability take into consideration other types of risks and hazards (not only natural)? And I miss some clear recommendations on what could be the best way forward for evolving the survey in the context of Nepal. Having done the first leg of the work, which other sources and local champions should be involved in Nepal to improve the tool?"

**(Reply to R1-11):** To enrich the discussion of the scalability of the methodology, we will add further information to Sections 4.6 ("Scalability to other data-scarce urban settings") and 4.7 ("Future work"), to include the following:

- Inclusion of additional hazard and risk types (see reply to R1-09).
- Suggestions for developing and applying the methodology in the Nepali context.
    - Potential local practitioners: non-governmental organisations (e.g., Nepal Mahila Ekata Samaj, NMES), national societies (e.g., National Society for Earthquake Technology – Nepal, NSET), research institutes (e.g., Southasia Institute of Advanced Studies, SIAS).
    - Further sources of information: government agency reports; preparedness plans across scales (e.g., national, regional, local, community levels); Nepali (and additional language) newspaper articles; social media content; further semi-structured interviews and focus group discussions with members of urban poor communities and practitioners working with them; etc.
    - Ensuring the dynamic nature of the tool through continued evaluation and implementation of recommendations (e.g., using Theory of Change).

**Reviewer 2 (R2) (Anonymous)**

**(R2-00):** "This study presents a methodology to identify single and multi-hazards using the Kathmandu Valley, Nepal, as case study. The methodology contains different approaches,

such as searching academic literature, grey literature, media, databases, and social media, but also involving directly stakeholders though workshops. It is important to have this type of data, especially for regions with less resources. I particularly liked the use of the multi-hazard data during the workshop to work on potential DRR strategies. I think the study is aligned with the scope of the journal. I have minimal technical remarks on the methods, but some on the discussions."

**(Reply to R2-00):** We would like to express our sincere thanks to Reviewer 2 for their time, insights and constructive engagement in refining the manuscript. We respond below to R2's main comments.

**(R2-01): "**The manuscript is well written but also very long. I believe part of that is due to repetitive language that could be avoided and made more direct. There are too many cross-references (ex: "This will be shown in section XX"; "… this was shown in section YY"). While I would not expect significant changes on the writing for this study, my recommendation for the authors on future work is to keep things direct, linear and to the point."

**(Reply to R2-01):** Thank you for your comment on the written style of the manuscript; we are grateful for this constructive feedback on how to present subsequent research more concisely. When writing this manuscript, we wanted to clearly signpost the reader and indicate linkages between sections to aid readability and ensure a logical narrative. In repeating some content across multiple sections, we aimed to outline important information found elsewhere in the text in the case that the reader only engages with some sections of the manuscript. We will go back over the manuscript and see if places where we used cross-referencing might be reduced without loss of functionality and will double check for any repetition to reduce text.

**(R2-02):** "Most crucially for me is that the discussion sections 4.1, 4.2 and 4.3 read as a summary or continuation of the results. Instead, the discussion should offer interpretation and reflection on the results (what do they mean?) and comparisons with past findings (literature)."

**(Reply to R2-02):** Thank you for drawing our attention to this area for refinement within the Discussion section. Please see responses to comments on Sections 4.1, 4.2 and 4.3 below for details of the in-text amendments we will make.

**Detailed comments:**

**(R2-03):** "Line 48 to line 52 seems to be redundant, maybe a copying mistake?"

**(Reply to R2-03):** Within the identified text (lines 48-52), we will remove the final sentence as follows:

"**Table 1** summarises six systematic studies compiling multi-hazard interrelationships on a regional scale, including the systematic methodologies used and the region to which the methodology is applied. The methodologies vary from critical literature reviews to multi-hazard risk analyses as tools to gather information about multi-hazard interrelationships across geographical regions.  [Removed]"

**(R2-04):** "Line 78-79: Maybe briefly define technical terms, such as hazardscapes (line 78) and (Multi-hazard) interrelationship (line 79)."

**(Reply to R2-04):** We will define hazardscape as follows:

Hazardscape – "Here we use the framework of hazardscapes to understand the connections between hazards, physical landscapes, socio-political factors and global influences (e.g., Mustafa, 2005; Khan, 2009)."

We will add a definition of multi-hazard interrelationships and include examples on line 79 (see also reply to R1-02).

**(R2-05):** "Section 3.2: It is an interesting demonstration of the multi-hazard table (Figure 6) for designing causal diagrams. While not the aim of this study, the actual representation of these causal diagrams should include also direct effects. This increases the complexity of the multi-hazard events & impacts. In the examples provided, a storm also directly causes flood and extreme temperature also directly contributes to urban fires, and they should be included in the diagram."

**(Reply to R2-05):** Thank you for this comment. There are a number of secondary hazards we could identify on the left-hand side matrix of **Fig. 6** that are directly triggered by the primary hazards in the matrix, and which could then be placed into the causal diagram on the right-hand side. For example, a storm can cause (visible portion shown of the matrix on the left-hand side of **Fig. 6**) an earthquake, volcanic eruption, landslide, flood and ground collapse. Similarly, the extreme temperature (heat) can trigger or increase the probability of multiple other secondary hazards. The purpose of the causal diagrams is to give a simplified and illustrative diagram for the stakeholder to understand. To enforce that the causal diagrams are just one of many possibilities, we suggest leaving the diagram as is (so as not to overcomplicate it) and instead we will address in the text that the dynamic scenarios (causal diagrams) given on the right-hand side of **Fig. 6** are just one of many hazard interrelationship scenarios that can be derived from the matrix, and give some other examples in the text (including those that R2 mentions).

**(R2-06):** "Section 4.1: I do not follow what is meant by frequency magnitude relationship. In fact, I did not see a discussion on the frequency, but a quick description of some hazards that happened in the past. This is not informative. I think it's important to address: what do those numbers suggest? Why are some hazards more common than others? What should a stakeholder or reader take out from your analysis on the different hazards?"

**(Reply to R2-06):** In Section 4.1 ("Frequency magnitude information") we will refer to some of the studies which have summarised frequency magnitude relationships for multi-hazard interrelationships (e.g., Tilloy et al., 2019) and amend this section to include a discussion of the implications of these frequency magnitude relationships. We will also discuss why some hazards occur, or influence, Kathmandu Valley more frequently than others.

**(R2-07):** "Section 4.2: Your two challenges: "Globally, there is a focus on reporting and describing single hazards instead of detailed information on hazard interrelationships. // Globally, research and understanding of single hazards are more established than for multi-hazard interrelationships."

These are obvious remarks, and should be the starting point of the discussion and not a conclusive remark. The following sentences are just continuations of the results, mentioning how much low cases of multi-hazards are found. That is a result, not a discussion. You should aim for answering questions like: What does the literature say about it (more specifically: is this more common in the global south? Are there regions where this is not an issue? What have other studies done to overcome this?); Why is it harder to find detailed information on hazard interrelationships? How do your findings fit in the general scheme built in the literature?"

**(Reply to R2-07):** We thank the reviewer for this thoughtful comment and will take this on board. We will consider how to refine and develop the content of Section 4.2 ("Challenges in finding case study evidence of multi-hazard interrelationships that have influenced Kathmandu Valley") in response to the discussion points suggested. This will include added depth to the current manuscript on the following points:

- Examination of the literature on the sparseness of multi-hazard interrelationship information (e.g., in the context of the "Global South" compared to other regions).
- Some of the barriers that exist to the collection and documentation of multi-hazard interrelationships across contexts, and briefly mention some tentative ways that the disaster risk reduction (DRR) community might overcome these challenges.
- How the findings of the paper fit and are situated within the broader literature.

**(R2-08):** "Section 4.3: Even though there is more contextualization in this section, I think it is still a long section describing results, instead of really discussing them. A direct paragraph discussing the implications of the findings without describing results would be an improvement."

**(Reply to R2-08):** We will refine Section 4.3 ("Workshop findings") to further develop the contextualisation of the multi-hazard interrelationship scenarios and impact examples that we extracted from the respective Padlet pages. This will include implications of these data and their application in the context of existing disaster risk reduction (DRR) strategies and policy in Kathmandu Valley and Nepal more broadly.

References Cited in Our Reply **(* and bold = new reference)**

[revised manuscript text omitted]

---

## Author Response (AR1)

**Reply to reviewer comments on NHESS-2024-101**

We express our thanks to both reviewers (R1 and R2) for their time, insights and constructive engagement with our manuscript **"A methodology to compile multi-hazard interrelationships in a data-scarce setting: an application to Kathmandu Valley, Nepal"** by Harriet E. Thompson, Joel C. Gill, Robert Šakić Trogrlić, Faith E. Taylor, and Bruce D. Malamud, submitted on 28 May 2024 to *Natural Hazards and Earth System Sciences* (Manuscript NHESS-2024-101).

Below, we provide detailed responses to the reviewers' comments following the editor's invitation to submit minor revisions. We have also submitted a track change document comparing the original and revised manuscripts. Unless otherwise stated, line numbers refer to those in the original version of the manuscript, and we have added additional references to the reference list.

We believe that the amendments we have made in response to the reviewers' comments have improved the quality and argument of the manuscript and thank the reviewers again for their contributions.

**Reviewer 1 (R1) (Nathan Clark)**

**(R1-00):** "The paper provides a very interesting and timely methodological approach for identifying and compiling information on interrelationships among single and multi-natural hazards. Overall the paper is well written and well positioned for a place within the Special issue: Methodological innovations for the analysis and management of compound risk and multi-risk, including climate-related and geophysical hazards. Some minor considerations for the authors:"

**(Reply to R1-00)**: We thank R1 for their comments on the interesting and timely approach for compiling information. Below we respond to each comment in turn.

**Introduction:**

**(R1-01):** "Overall this section is clear and concise, however, a few parts of the narrative could flow better and need clarification.

In terms of data-scarcity on hazard impact data, and this quote: "One of the significant challenges facing urban areas in Nepal is the scarcity of hazard impact data, which is a barrier to effective DRR strategies (Chatterjee et al., 2015; SIAS, 2016)." The first thought that came to my mind is wondering the extent which this has changed almost 10 years since the 2015 Earthquake and from the sources referenced. And if we are talking about scarcity of multi-hazard impact data, I wonder how Kathmandu differs so much from other urban settings, even in the West."

**(Reply to R1-01):** The quote above (R1-01) has been removed.

We have added the following text to **Sect. 1** in the revised manuscript:

"In the Nepali context, most previous research and building of databases by academics, government organisations and non-governmental organisations (NGOs) has centred on the impacts of single hazards (Bhatta and Adhikari, 2024). There is an increasing shift from single hazard to multi-hazard approaches, exemplified by studies such as Khatakho et al. (2021) multi-layer risk assessment in the Kathmandu Valley that superimposed earthquake, fire, flood

and landslide risk. In response to recent multi-hazard events such as the Melamchi debris flow in 2021, part of a hazard cascade that caused multi-sectoral impacts on a regional scale across one year (Sharma et al., 2023), more research is focused on multi-hazard impact and risk assessments (e.g., Dunant et al., 2024). Examples of the breadth of natural hazard events in Nepal and subsequent cascading hazards or impacts include the following:

- The high-profile disaster of the Gorkha earthquake in April 2015 caused devastating impacts in Kathmandu Valley and beyond (Takai et al., 2016; Khatakho et al., 2021), with subsequent landslides on the periphery of the valley exacerbating these effects and prolonging the recovery effort (Kargel et al., 2015).

- Urban fires occur frequently and spread rapidly in areas of Kathmandu Valley with high population density (Khatakho et al., 2021), particularly in informal settlements where "marginalised" communities experience disproportionate hazard impacts and may have lower capacity to prepare and respond to hazard events (Brown et al., 2019; Dodman et al., 2022).

- Both fluvial and pluvial flooding are frequent in Kathmandu Valley during the mid-June to early September monsoon season. For example, during floods in early September 2021, heavy rainfall caused severe inundation across large areas of the valley and displaced hundreds of families in the Banshi Ghat area (Chaulagain et al., 2023).

These earthquake, fire, flood and landslide multi-hazard events exemplify the complexity of the interrelationships between hazards and their impacts and highlight the need to understand how these events relate to the geographical contexts in which they occur.

**(R1-02):** "On lines 84 and 95 the authors note the Kathmandu experiences many single and multi-hazards events without qualifying it. Maybe the examples under 110-120 could be brought up under points 84 and 95 to create a more coherent narrative."

**(Reply to R1-02):** Examples of natural hazard events and cascades of further hazards or impacts in Nepal have been moved from L110-L120 (original manuscript) to L112-123 (revised manuscript).

**(R1-03):** "From lines 95 down, it is not immediate clear why the authors are highlighting the squatter settlements, and even including a visual. Is it only to illustrate population growth and growing urbanisation, or is it related to the heightened risks and impacts on vulnerable groups? This is connected indirectly to the Urban Fire bullet point, and then again later made clearer in the Results and Discussions sections with the focus on impacts to marginalised communities. I suggest the authors make these connections more clear in the Introduction when introducing the section on squatter settlements."

**(Reply to R1-03):** We have expanded the paragraph which introduces squatter settlements in Kathmandu Valley. This paragraph includes an explanation for the increased vulnerability and disproportionate burden of impacts experienced by members of "marginalised" groups. Please see as follows the added text:

"Within this population, "marginalised" communities, including residents of squatter settlements within the broader grouping of urban poor communities, experience a disproportionate burden of hazard impacts due to their heightened socio-economic vulnerability (Pelling et al., 2004; Gorman-Murray et al., 2018; Dodman et al., 2022). In mainstream hazard impact data, more vulnerable groups often lack representation (Osuteye et al. 2017). Addressing this data gap is crucial to prevent DRR strategies from unintentionally

exacerbating existing social inequalities (Brown et al., 2019). To illustrate the spatial distribution of socially marginalised communities in one region of Kathmandu Valley, **Fig. 1** presents six squatter settlements, with a total population of 7270 people in 2019 (Khanal and Khanal, 2022), out of 53 squatter settlements that are documented in the valley (DUDBC, 2010). Within Kathmandu Valley, 35 of 53 (66%) squatter settlements are located along the banks of significant river corridors like Bagmati (**Fig. 1**). Kathmandu Valley experiences a breadth of single natural hazard types (Pradhan et al., 2020; Whitworth et al., 2020; Khatakho et al., 2021), with the potential for interrelationships to occur between these hazards and across varying spatial and temporal scales."

**Methodology:**

**(R1-04):** "Only using English sources is certainly one of the main weaknesses of the study. But it is very good that the authors have recognised and highlighted the limitation in a number of places within the document.

What remains unclear is what is meant by "grey literature". Perhaps I missed it, but I find only one small explanation on line 608 which references "e.g., UNDRR reports". Did the authors include in their literature search any other NGO/IGO and government agency reports, and perhaps preparedness plans at national/regional/local/community levels? They could likely find more hazard specific frequency magnitude/impact information through specific agencies in the regions, for instance NSET for geophysical hazards. https://www.nset.org.np/nset2012/. These types of sources seem very relevant to have been included considering the scope of the paper,"

**(Reply to R1-04):** We have included Auger's (1998) widely accepted definition of grey literature. In addition, we have listed further subtypes (on lines 205-206) of grey literature that we selected for inclusion in the single hazard and multi-hazard interrelationships database.

**(R1-05):** "Line 214 notes: "We used the methodology described in the literature review in Sect. 2.1 to conduct Boolean searches for single hazards and multi-hazard interrelationships that might occur in Kathmandu Valley." Did this also include events which have already occurred? I suggest to make that clear."

**(Reply to R1-05):** We have amended this sentence to read: "We used the methodology described in the literature review in **Sect. 2.1** to conduct Boolean searches for potential single hazard types and multi-hazard interrelationships that have influenced or could potentially influence Kathmandu Valley.".

**(R1-06):** "I appreciate the thought behind pulling out the column headings into separate boxes for ease of reading in Figures 2 and 3, but I think it might create even more difficulty for the reader. Or at least it makes the presentation of the Figures look quite messy. Consider if those extra boxes are really needed. Perhaps instead just highlight the relevant text in different colours in the tables."

**(Reply to R1-06):** Thank you for this comment; we have discussed and experimented with this suggested visualisation to improve the clarity of **Figures 2** and **3.** We have removed the separate boxes detailing the column headers in both figures to improve presentation and ease of viewing for the reader.

**(R1-07):** "Line 293 notes: "These participants were selected based on their in-depth knowledge of the Kathmandu Valley context and existing connections built on pre-established working relationships." What does this say about their specific backgrounds and expertise contributing to this topic? Especially given the small number or participants, this clarification is needed in the methodology section. I found the explanation (in part) only at the end of the document in the Limitations section."

**(Reply to R1-07):** To provide clarification, we have added the following sentences to this section (**Sect. 2.4**) in line with participants' responses to the pre-workshop consent form to give more detail about the specific backgrounds and expertise contributing to this topic

"The virtual Teams workshop aimed to co-produce multi-hazard interrelationship scenarios and their impacts through two workshop activities. To minimise the potential effect of power asymmetries (Secor, 2010; Wolf, 2018), we balanced the number of Nepali or Nepal-based (four) and British or UK-based (three) participants, and female (two) and male (five) participants to support participants in feeling comfortable to share their knowledge and perspectives. These participants were selected based on their in-depth knowledge of single hazards and multi-hazard interrelationships in the Kathmandu Valley context, as well as existing connections built on pre-established working relationships (Wilmsen, 2008). Participants were drawn from the non-governmental organisation (NGO) or international non-governmental organisation (INGO), national society, research institute, and academic sectors. The research backgrounds of participants were social scientists, physical scientists (e.g., Geography), and interdisciplinary scientists (e.g., Thematic Lead: Climate and Resilience) and ranged from early career (e.g., Research Associate) to senior career (e.g., Professor). We utilised the snowball sampling technique (Secor, 2010) to encourage participants to suggest any further colleagues who they thought might be interested in participating in the same workshop for us to contact. All participants gave informed consent for participation, indicating their requested level of anonymity (i.e., any combination or none of the following: full name, position, institution).".

**Results:**

**(R1-08):** "Figure 5 is great! It also made me quite curious about some of the relationships. For instance I had to look up how a hailstorm can trigger a volcanic eruption.

Line 407 notes: "Additionally, we shared Fig. 6 with workshop participants to illustrate the value of the multi-hazard interrelationship matrix in extracting relevant multi-hazard scenarios." It is not clear if they were asked for feedback on that Figure, or if it was also used as a tool in the workshop? E.g. What did they think of it? Was it useful?"

**(Reply to R1-08):** Although it was not within the workshop's scope to ask for participant feedback on the multi-hazard interrelationships matrix itself (other than introducing them to it), we did use **Fig. 6** as a tool for participants to explore potential multi-hazard scenarios that could influence Kathmandu Valley. We have added the sentences as follows:

"We shared **Fig. 6** with workshop participants to illustrate the value of the multi-hazard interrelationship matrix in extracting relevant multi-hazard scenarios. **Figure 6** was then made available virtually to participants during workshop discussions and acted as a visualisation tool and talking point to explore further multi-hazard interrelationships that have influenced or could

influence Kathmandu Valley beyond those archetypal examples that participants may associate with the region.".

**(R1-09):** "Line 462 notes: "Including anthropogenic hazards and related processes could form the basis of future developments of this work." That seems like a fairly significant outcome and should probably be highlighted in sections 4.6 and 4.7 on future work in this direction.""

**(Reply to R1-09):** We agree with the reviewer that this should be highlighted as it is a part of our future research. Following discussions regarding how to expand upon this sentence in **Sect. 4.6 Future work**, we have added the following paragraphs:

"Additionally, including anthropogenic processes within the methodology can add nuance to the breadth of single hazard types and multi-hazard interrelationships across geographical contexts. Here, we understand anthropogenic processes as "intentional, non-malicious human activities" as defined by Gill and Malamud (2017). In their research, Gill and Malamud (2017) present a systematic literature review of past research on anthropogenic processes, focusing on categorising artificial ground and land use. Following this, they characterise the role of anthropogenic processes in triggering natural hazard events and catalysing or impeding the interrelationships between natural hazards (Gill and Malamud, 2017). Moving forward, we recognise the need to develop our methodology further to include anthropogenic processes as an integral component of any hazardscape.

Another potential expansion is the inclusion of non-natural hazards, including the following clusters as defined in the UNDRR-ISC Hazard Information Profiles (Murray et al., 2021):

- Biological: a broad range of hazards of "organic origin".
- Chemical: human exposure to chemicals of human and natural origin.
- Environmental: degradation of "natural systems and ecosystem services".
- Societal: "human activities and choices" that present risks to communities and environments.
- Technological: failure of existing and emerging technology, impacting both within and outside systems.

Although our methodology has focused on natural hazard types, it can be applied to a broader dataset, including additional hazard types and spatial and temporal scales, to capture hazardscapes across different geographical contexts. Natural hazards are situated within the wider systems in which they occur and can be directly or indirectly influenced by anthropogenic processes (Gill and Malamud, 2017), particularly in urban contexts. We aspire that integrating anthropogenic processes and additional hazard types into the existing methodology and multi-hazard interrelationship matrix results in a more nuanced understanding of urban hazardscapes. These elements will extend existing multi-hazard scenarios to incorporate risk variables. This knowledge is significant for practitioner stakeholders working in specific urban contexts to evaluate which components of dynamic risk scenarios can be targeted to reduce impacts on at-risk and "marginalised" communities.".

**Discussion:**

**(R1-10):** "In section 4.3.2. It could be interesting for the authors to reflect on HOW the different types of variables related to vulnerabilities and impacts (also within different spatial and

temporal scales) could be considered within such a matrix. The authors do provide some considerations for this in sections 4.6 and 4.7, maybe there is a way to also provide a few of these reflections already in 4.3.2."

**(Reply to R1-10):** Due to the restructuring of the discussion, **Sect. 4.3.2 Impact examples** is now **Sect. 4.2.2 Impact examples**. Within **Sect. 4.2.2** we have examined potential approaches of incorporating variables of vulnerability and impacts into the existing multi-hazard interrelationship matrix. To respond to comment **R1-10**, we have included the following paragraphs:

"Despite increasing focus on cascading and disaggregated impacts, there remain gaps in multi-hazard interrelationship knowledge, including a detailed understanding of the direct and indirect impacts of multi-hazards necessary for effective mitigation (Šakić Trogrlić et al., 2024a). A potential extension of the workshop with practitioner stakeholders is to incorporate questions that consider variables of vulnerability and impact into the multi-hazard interrelationship matrix. Existing multi-hazard visualisations (e.g., Gustafsson et al., 2023) illustrate the interrelationships and impacts resulting from hazard cascades and provide potential approaches to incorporating these variables within our methodology. For example, Sharma et al. (2023) illustrate hazard cascades and their impacts in the central Himalayas, including the duration, scale and sector influenced by each hazard cascade event using a multi-hazard interrelationship matrix similar to Fig. 5 in our paper.

Another important consideration in visualising the multi-hazard interrelationship matrix is the useability of the tool by practitioner stakeholders engaged in the hazardscape region. When incorporating the tool into existing DRR strategies, understanding the spatial and temporal components of multi-hazard events is critical in coordinating an appropriate and tailored response. For example, De Angeli et al. (2022) developed a multi-hazard risk framework for spatial-temporal impact analysis and applied it to a seismic and flood damage scenario in the Po Valley, Italy. The spatial and temporal evolution of the multi-hazard event scenario is visualised with the following components: hazard maps at various time instants, the temporal evolution of hazard impacts, and the impacted area.

Both Sharma et al. (2023) and De Angeli et al.'s (2022) visualisations present potential approaches to expanding our existing multi-hazard interrelationship matrix in the Kathmandu Valley context. Including impact variables within Fig. 5 would enhance our methodology's scalability and utility within DRR strategies (discussed further in Sect. 4.5 and Sect. 4.6). A clear visualisation of the evolution of impacts across type and spatial-temporal extent in a figure or series of figures would be helpful as a dissemination tool in decision-making processes.

**(R1-11):** "4.6 and 4.7 are really nice to read. Two things that do however seem to be missing (or not explicitly enough) in terms of scalability of the matrix: should scalability take into consideration other types of risks and hazards (not only natural)? And I miss some clear recommendations on what could be the best way forward for evolving the survey in the context of Nepal. Having done the first leg of the work, which other sources and local champions should be involved in Nepal to improve the tool?"

**(Reply to R1-11):** To enrich the discussion of the scalability of the methodology, we have added further information to **Sect. 4.5 Scalability to other data-scarce urban settings** and **Sect. 4.6 Future work**, as follows:

**Section 4.5**

[revised manuscript text omitted]

**Reviewer 2 (R2) (Anonymous)**

**(R2-00):** "This study presents a methodology to identify single and multi-hazards using the Kathmandu Valley, Nepal, as case study. The methodology contains different approaches, such as searching academic literature, grey literature, media, databases, and social media, but also involving directly stakeholders though workshops. It is important to have this type of data, especially for regions with less resources. I particularly liked the use of the multi-hazard data during the workshop to work on potential DRR strategies. I think the study is aligned with the scope of the journal. I have minimal technical remarks on the methods, but some on the discussions."

**(Reply to R2-00):** We would like to express our sincere thanks to Reviewer 2 for their time, insights and constructive engagement in refining the manuscript. We respond below to R2's main comments.

**(R2-01):** "The manuscript is well written but also very long. I believe part of that is due to repetitive language that could be avoided and made more direct. There are too many cross-references (ex: "This will be shown in section XX"; "… this was shown in section YY"). While I would not expect significant changes on the writing for this study, my recommendation for the authors on future work is to keep things direct, linear and to the point."

**(Reply to R2-01):** Thank you for your comment on the written style of the manuscript; we are grateful for this constructive feedback on how to present subsequent research more concisely. When writing this manuscript, we wanted to clearly signpost the reader and indicate linkages between sections to aid readability and ensure a logical narrative. In repeating some content across multiple sections, we aimed to outline important information found elsewhere in the text in the case that the reader only engages with some sections of the manuscript. We have reviewed the manuscript, removing repeated content whilst retaining some of the signposting across sections.

**(R2-02):** "Most crucially for me is that the discussion sections 4.1, 4.2 and 4.3 read as a summary or continuation of the results. Instead, the discussion should offer interpretation and reflection on the results (what do they mean?) and comparisons with past findings (literature)."

**(Reply to R2-02):** Thank you for drawing our attention to this area for refinement within **Sect. 4 Discussion**. Please see our responses (below) to your comment for details of our in-text amendments.

**Detailed comments:**

**(R2-03):** "Line 48 to line 52 seems to be redundant, maybe a copying mistake?"

**(Reply to R2-03):** Within the identified text, we have removed the final sentence as follows:

**"Table 1** summarises six studies compiling multi-hazard interrelationships on a regional scale, including the systematic methodologies used and the region to which the methodology is applied. The methodologies vary from critical literature reviews to multi-hazard risk analyses as tools to gather information about multi-hazard interrelationships across geographical regions.  [Removed]"

**(R2-04):** "Line 78-79: Maybe briefly define technical terms, such as hazardscapes (line 78) and (Multi-hazard) interrelationship (line 79)."

**(Reply to R2-04):** We have defined hazardscape as follows:

Hazardscape – "applied in this paper as a framework to understand the connections between hazards, physical landscapes, socio-political factors and global influences (e.g., Mustafa, 2005; Khan, 2009)…"

We have added a definition of multi-hazard interrelationships and included examples on lines 42-53 in the amended manuscript as follows:

Multi-hazard interrelationship – "Different authors have developed classifications for multi-hazard interrelationships (e.g., Kappes et al., 2010; Duncan, 2014; Van Westen et al., 2014). However, each classification shares many features and typically includes one or more of the following interrelationship types (Gill and Malamud, 2016, 2017; Ciurean et al., 2018; Gill et al., 2022):

- Compound (or coincident hazards): two or more independent hazards affect the same area spatially and/or temporally.

- Concurrent or consecutive hazards: two or more hazards occur consecutively, resulting in increased stress on a certain area.
    - Triggering relationships: one hazard causes another hazard to occur.
    - Increased probability relationships: one hazard increases the magnitude and/or likelihood of further hazards in the future.

- Catalysis/impedance relationships: the action of a primary hazard triggers or increases the probability of a secondary hazard.".

**(R2-05):** "Section 3.2: It is an interesting demonstration of the multi-hazard table (Figure 6) for designing causal diagrams. While not the aim of this study, the actual representation of these causal diagrams should include also direct effects. This increases the complexity of the multihazard events & impacts. In the examples provided, a storm also directly causes flood and extreme temperature also directly contributes to urban fires, and they should be included in the diagram."

**(Reply to R2-05):** Thank you for this comment. There are a number of secondary hazards we could identify on the left-hand side matrix of **Fig. 6** that are directly triggered by the primary hazards in the matrix, and which could then be placed into the causal diagram on the right-hand side. For example, a storm can cause (visible portion shown of the matrix on the left-hand side of **Fig. 6**) an earthquake, volcanic eruption, landslide, flood and ground collapse. Similarly, the extreme temperature (heat) can trigger or increase the probability of multiple other secondary hazards. The purpose of the causal diagrams is to give a simplified and illustrative diagram for the stakeholder to understand. To enforce that the causal diagrams are just one of many possibilities, we have left the diagram as is (so as not to overcomplicate it) and instead we have addressed in the text (lines 473–477 in the amended manuscript) that the dynamic scenarios (causal diagrams) given on the right-hand side of **Fig. 6** are just one of many hazard interrelationship scenarios that can be derived from the matrix. We have given the examples that R2 has mentioned in the text.

**(R2-06):** "Section 4.1: I do not follow what is meant by frequency magnitude relationship. In fact, I did not see a discussion on the frequency, but a quick description of some hazards that happened in the past. This is not informative. I think it's important to address: what do those numbers suggest? Why are some hazards more common than others? What should a stakeholder or reader take out from your analysis on the different hazards?"

**(Reply to R2-06):** We have removed **Section 4.1** in the original manuscript and have converted the contents (i.e., "major event typical frequency" information) into a new **Table 3** in the results **Sect. 3.1 Single hazard types influencing Kathmandu Valley**. Please see the newly inserted **Table 3** and text below:

"Across the single hazard types, those that occur most frequently (e.g., flood, urban fire) or recently occurred with higher magnitudes (e.g., earthquake), were the most common hazard type reported by different sources, and the impacts of the hazard were often described in more detail. Conversely, it was more challenging to find evidence for hazard types where (a) major events occur less frequently (or have no direct evidence of occurrence, i.e., may only be theoretically possible in Kathmandu Valley), such as volcanic eruption or impact events or (b) where typical 'major' events have localised impacts and are considered an every-day occurrence by the local population such as soil subsidence. These limitations are explored further in **Sect. 4.4.**

As illustrated by the results for single hazard types, Kathmandu Valley is exposed to a plethora of hazard types, notably earthquake, urban fire, flood and landslide (Gautam et al., 2021; Khatakho et al., 2021), owing to factors such as its tectonic location, high building density, position in the wider Bagmati River Basin, incidence of the annual monsoon, and steep topography. In extracting major event typical frequency information from our analysis, shown in **Table 3**, we aim to give a preliminary indication of the prevalence and size of specific hazard types where this information is available, especially for hazard types that may be overlooked as a risk to Kathmandu Valley.

**Table 3. Summary of major event typical frequencies for four single hazard types occurring in Kathmandu Valley or Nepal.**

| Hazard | Major events, including date and magnitude | Typical frequency of major events |
| --- | --- | --- |

| | | |
|---|---|---|
| Earthquake | • 1255 (magnitude unknown), 1344 (magnitude unknown), 1833 (Mw ~7.7), 1934 (Mw 8.2) and April 2015 (Mw 7.8) (Rajendran, 2021). | • At least one major earthquake each century in Nepal (Tiwari and Paudyal, 2024).
• Magnitude Mw 5.0-6.5 earthquakes in Nepal have 5-10 years mean return period (Sharma and Biswas, 2024). |
| Volcanic eruption | • 20 Volcanic Explosivity Index (VEI) 6-8 eruptions were dated between 1.2 Ma to 1991 AD in Southeast Asia (De Maisonneuve and Bergal-Kuvikas, 2020). | • Probabilities of VEI 6, 7 and 8 volcanic eruptions occurring somewhere in Southeast Asia in 10 years are ~0.15, ~0.012 and ~0.001 (Whelley et al. 2015). |
| Flood | • "Frequent" flooding during the annual monsoon season (magnitudes not mentioned) (e.g., Chaudhary et al., 2024; Danegulu et al., 2024). | • Daily maximum floods for 5, 10 and 25-year return periods were estimated as 876, 1077, and 1331 $m^3$ $s^{-1}$ under present climatic conditions (Mishra et al. 2024). |
| Tornado | • Windstorm Parvana was the first recorded tornado in Nepal (mean speed: 250 km $h^{-1}$; estimated size: 200 $km^2$) (Chhetri et al., 2019). | • Windstorm Parvana was the largest-scale storm in over seventy years (Gautam et al., 2020). |

Taking the major event typical frequencies presented in **Table 3** as exemplars rather than an exhaustive compilation, these records suggest that specific hazards within the geophysical, hydrological, and atmospheric hazard groups have the most quantitative information on major event typical frequencies in the context of Kathmandu Valley. The remaining hazard groups – shallow Earth processes, biophysical and space/celestial – have some major event typical frequency information for some hazard types influencing Kathmandu Valley, but this is typically limited to qualitative descriptions.".

**(R2-07):** "Section 4.2: Your two challenges: "Globally, there is a focus on reporting and describing single hazards instead of detailed information on hazard interrelationships. // Globally, research and understanding of single hazards are more established than for multi-hazard interrelationships."

These are obvious remarks, and should be the starting point of the discussion and not a conclusive remark. The following sentences are just continuations of the results, mentioning how much low cases of multi-hazards are found. That is a result, not a discussion. You should aim for answering questions like: What does the literature say about it (more specifically: is this more common in the global south? Are there regions where this is not an issue? What have other studies done to overcome this?); Why is it harder to find detailed information on hazard interrelationships? How do your findings fit in the general scheme built in the literature?"

**(Reply to R2-07):** We thank the reviewer for this thoughtful comment and have taken this on board. We have extensively refined and developed the content of **Section 4.1 Challenges in**

**finding case study evidence of multi-hazard interrelationships that have influenced Kathmandu Valley** in response to the discussion points suggested.

Please see the following additions:

[revised manuscript text omitted]

**(R2-08):** "Section 4.3: Even though there is more contextualisation in this section, I think it is still a long section describing results, instead of really discussing them. A direct paragraph discussing the implications of the findings without describing results would be an improvement."

**(Reply to R2-08):** We have refined **Section 4.2 Workshop findings** to further develop the contextualisation of the multi-hazard interrelationship scenarios and impact examples that we extracted from the respective Padlet pages. This includes the implications of these data and their application in the context of existing disaster risk reduction (DRR) strategies and policy in Kathmandu Valley and Nepal more broadly and bringing in more literature.

Please see these additions as follows:

**4.2.1. Multi-hazard interrelationship scenarios**

"Facilitating workshops to better understand multi-hazard cascades can be valuable for practitioner stakeholders considering emerging risks and future scenarios to inform decision-making processes (e.g., Riddell et al., 2019; Strong et al., 2020). Developing multi-hazard scenarios from workshop participants and the multi-hazard interrelationship matrix (**Fig. 5**) can be helpful for hazard practitioners and agencies working in the DRR space. Applications include evaluating the effectiveness of preparedness and response systems, guiding land-use planning, communicating educational messages towards at-risk communities, and facilitating dialogue between practitioner stakeholders and at-risk communities (Gill et al., 2020). In the Nepali context, developing and quantifying multi-hazard scenarios would support preparedness and recovery strategies and the allocation of resources on provincial and national scales (Gautam et al., 2021).".

**4.2.2 Impact examples**

"Despite increasing focus on cascading and disaggregated impacts, there remain gaps in multi-hazard interrelationship knowledge, including a detailed understanding of the direct and indirect impacts of multi-hazards necessary for effective mitigation (Šakić Trogrlić et al., 2024a). A potential extension of the workshop with practitioner stakeholders is to incorporate questions that consider variables of vulnerability and impact into the multi-hazard

interrelationship matrix. Existing multi-hazard visualisations (e.g., Gustafsson et al., 2023) illustrate the interrelationships and impacts resulting from hazard cascades and provide potential approaches to incorporating these variables within our methodology. For example, Sharma et al. (2023) illustrate hazard cascades and their impacts in the central Himalayas, including the duration, scale and sector influenced by each hazard cascade event using a multi-hazard interrelationship matrix similar to **Fig. 5** in our paper.

Another important consideration in visualising the multi-hazard interrelationship matrix is the useability of the tool by practitioner stakeholders engaged in the hazardscape region. When incorporating the tool into existing DRR strategies, understanding the spatial and temporal components of multi-hazard events is critical in coordinating an appropriate and tailored response. For example, De Angeli et al. (2022) developed a multi-hazard risk framework for spatial-temporal impact analysis and applied it to a seismic and flood damage scenario in the Po Valley, Italy. The spatial and temporal evolution of the multi-hazard event scenario is visualised with the following components: hazard maps at various time instants, the temporal evolution of hazard impacts, and the impacted area.

Both Sharma et al. (2023) and De Angeli et al.'s (2022) visualisations present potential approaches to expanding our existing multi-hazard interrelationship matrix in the Kathmandu Valley context. Including impact variables within **Fig. 5** would enhance our methodology's scalability and utility within DRR strategies (discussed further in **Sect. 4.5** and **Sect. 4.6**). A clear visualisation of the evolution of impacts across type and spatial-temporal extent in a figure or series of figures would be helpful as a dissemination tool in decision-making processes.".

References Cited in Our Reply **(\* and bold = new reference)**
New references added to the Reference list in the amended manuscript

**\*Boaz, A. and Hayden, C.: Pro-active evaluators: enabling research to be useful, usable and used, Evaluation, 8(4), 440-453, https://doi.org/10.1177/13563890260620630, 2002.**

**\*Auger, C.P.: Information sources in grey literature (guides to information sources), 4, London, Bowker Saur, ISBN 3110977230, 1998.**

**\*Bhatta, S. and Adhikari, B.R.: Comprehensive risk evaluation in Rapti Valley, Nepal: A multi-hazard approach, Progress in Disaster Science, 23, 100346, https://doi.org/10.1016/j.pdisas.2024.100346, 2024.**

Brown, S., Budimir, M., Upadhyay Crawford, S., Clements, R., and Sneddon, A.: Gender and Age Inequality of Disaster Risk: Policy Brief, UNICEF and UN Women, 2019.

**\*Chaudhary, U., Shah, M.A.R., Shakya, B.M., and Aryal, A.: Flood Susceptibility and Risk Mapping of Kathmandu Valley Watershed, Nepal, Sustainability, 16(16), 7101, https://doi.org/10.3390/su16167101, 2024.**

Chaulagain, D., Rimal, P.R., Ngando, S.N., Nsafon, B.E.K., Suh, D., and Huh, J.S.: Flood susceptibility mapping of Kathmandu metropolitan city using GIS-based multi-criteria decision analysis, Ecol. Indic., 154, https://doi.org/10.1016/j.ecolind.2023.110653, 2023.

Chhetri, T.B., Dawadi, B., and Dhital, Y.P.: Detection of a Tornado Event on 31 March 2019 and Its Effects on the Eastern Part of Nepal, Indian Journal of Science and Technology, 12(38), https://doi.org/10.17485/ijst/2019/v12i38/146311, 2019.

**\*Ciurean, R., Gill, J., Reeves, H.J., O'Grady, S. and Aldridge, T.: Review of multi-hazards research and risk assessments, British Geological Survey, Nottingham, https://nora.nerc.ac.uk/id/eprint/524399, 2018.**

**\*Claassen, J.N., Ward, P.J., Daniell, J., Koks, E.E., Tiggeloven, T. and de Ruiter, M.C.: A new method to compile global multi-hazard event sets, Sci. Rep., 13(1), 13808, https://doi.org/10.1038/s41598-023-40400-5, 2023.**

**\*Danegulu, A., Karki, S., Bhattarai, P.K., and Pandey, V.P.: Characterizing urban flooding in the Kathmandu Valley, Nepal: the influence of urbanization and river encroachment, Nat. Hazards, 1-25, https://doi.org/10.1007/s11069-024-06650-w, 2024.**

De Angeli, S., Malamud, B.D., Rossi, L., Taylor, F.E., Trasforini, E., and Rudari, R.: A multi-hazard framework for spatialtemporal impact analysis, Int. J. Disast. Risk Re., 73, https://doi.org/10.1016/j.ijdrr.2022.102829, 2022.

**\*De Maisonneuve, C.B. and Bergal-Kuvikas, O.: Timing, magnitude and geochemistry of major Southeast Asian volcanic eruptions: identifying tephrochronologic markers, J. Quaternary Sci., 35(1-2), 272-287, https://doi.org/10.1002/jqs.3181, 2020.**

Department of Urban Development and Building Construction (DUDBC): Department of Urban Development and Building Construction's Report on Squatter Settlements in Kathmandu Valley, DUDBC: Kathmandu, Nepal, 2010.

Dodman, D., Hayward, B., Pelling, M., Castan Broto, V., Chow, W., Chu, E., Dawson, R., Khirfan, L., McPhearson, T., Prakash, A., Zheng, Y., and Ziervogel, G.: Cities, Settlements and Key Infrastructure, in: Climate Change 2022: Impacts, Adaptation and Vulnerability, Contribution of Working Group II to the Sixth Assessment Report of the Intergovernmental Panel on Climate Change, Cambridge University Press, Cambridge, UK and New York, NY, USA, 907–1040, https://doi.org/10.1017/9781009325844.008, 2022.

**\*Dunant, A., Robinson, T.R., Densmore, A.L., Rosser, N.J., Rajbhandari, R.M., Kincey, M., Li, S., Awasthi, P.R., Van Wyk de Vries, M., Guragain, R. and Harvey, E.: Impacts from cascading multi-hazards using hypergraphs: a case study from the 2015 Gorkha earthquake in Nepal, EGUsphere [preprint], 1-28, https://doi.org/10.5194/egusphere-2024-1374, 2024.**

**\*Duncan, M.J.: Multi-hazard assessments for disaster risk reduction: lessons from the Philippines and applications for non-govermental organisations, Ph.D. thesis, University College London, London, https://discovery.ucl.ac.uk/id/eprint/1452516, 2014.**

**\*Formetta, G. and Feyen, L.: Empirical evidence of declining global vulnerability to climate-related hazards, Global Environ. Chang., 57, 101920, https://doi.org/10.1016/j.gloenvcha.2019.05.004, 2019.**

[revised manuscript text omitted]